# Quantitative modelling of amino acid transport and homeostasis in mammalian cells

Gregory Gauthier-Coles [1], Jade Vennitti[1], Zhiduo Zhang[2], William C. Comb[3], Shuran Xing[3], Kiran Javed[1], Angelika Bröer[1] & Stefan Bröer [1✉]

Homeostasis is one of the fundamental concepts in physiology. Despite remarkable progress in our molecular understanding of amino acid transport, metabolism and signaling, it remains unclear by what mechanisms cytosolic amino acid concentrations are maintained. We propose that amino acid transporters are the primary determinants of intracellular amino acid levels. We show that a cell's endowment with amino acid transporters can be deconvoluted experimentally and used this data to computationally simulate amino acid translocation across the plasma membrane. Transport simulation generates cytosolic amino acid concentrations that are close to those observed in vitro. Perturbations of the system are replicated in silico and can be applied to systems where only transcriptomic data are available. This work explains amino acid homeostasis at the systems-level, through a combination of secondary active transporters, functionally acting as loaders, harmonizers and controller transporters to generate a stable equilibrium of all amino acid concentrations.

[1] Research School of Biology, Australian National University, Canberra, ACT, Australia. [2] Division of Genome Science and Cancer, ACRF INCITe Centre - ANU Node, John Curtin School of Medical Research, Australian National University, Canberra, ACT, Australia. [3] Axcella Health Inc., Cambridge, MA, USA. ✉email: Stefan.broer@anu.edu.au

The concept of homeostasis is a cornerstone of physiological research. Similar to other metabolites, plasma and cellular amino acid concentrations are kept within their physiological range. After a meal, amino acid concentrations rise, but return to homeostatic levels within hours[1,2]. In contrast to glucose, which is found at lower than plasma concentrations in the cytosol[3], cytosolic amino acid levels are several-fold higher than those in blood plasma[4]. This provides sufficient amino acid pools to support protein biosynthesis and contributes to the osmotic pressure of the cytosol[5].

Two signaling systems regulate amino acid homeostasis, namely, mTORC1 (mechanistic Target of Rapamycin Complex 1) and GCN2/ATF4 (General control nonderepressible 2/Activating transcription factor 4). When cytosolic and lysosomal amino acid levels are sufficient, they activate the mTORC1 complex[6], needed for cell division, efficient translation, and ribosome production[7]. Depletion of amino acids switches off mTORC1, but also generates a more specific response to deal with amino acid imbalances via the GCN2/ATF4 pathway of the integrated stress response[8].

Amino acid transport in mammalian cells is mediated by >60 different secondary active transporters, namely, uniporters, symporters, and antiporters (Fig. 1)[4,9]. Uniporters for amino acids do not occur frequently in mammalian cells, because they equilibrate cytosolic and plasma amino acid concentrations, thus preventing elevated levels of amino acids in the cytosol. Exceptions are cationic amino acid transporters, such as CAT1 (SLC7A1)[10], which accumulate their substrates due to a positive charge. Examples of widely expressed symporters are the Na+-neutral AA cotransporters SNAT1 (SLC38A1) and SNAT2 (SLC38A2)[5,11]. Another widely expressed group of amino acid transporters are antiporters[12]. Examples are LAT1 (SLC7A5, large neutral amino acids), y+LAT2 (SLC7A6, neutral and cationic AA), ASCT1 (SLC1A4, small neutral AA), and ASCT2 (SLC1A5, polar neutral AA)[13]. Although antiporters cannot mediate net uptake, they are upregulated in many fast-growing cell types, especially cancer cells[13]. Cationic amino acids can enter cells through uniporters, such as CAT1, or through antiporters, such as y+LAT2, which exchanges a neutral amino acid together with Na+ or K+ against a cationic amino acid[14]. Cysteine is acquired by cells as a neutral amino acid through symporters, but can also be acquired as negatively charged cystine (Cys–S–S–Cys) in exchange for glutamate through the antiporter xCT (SLC7A11)[15]. Some of these transporters are actively regulated by ATF4/GCN2 and mTORC1, such as CAT1 (SLC7A1), xCT, and SNAT2[16].

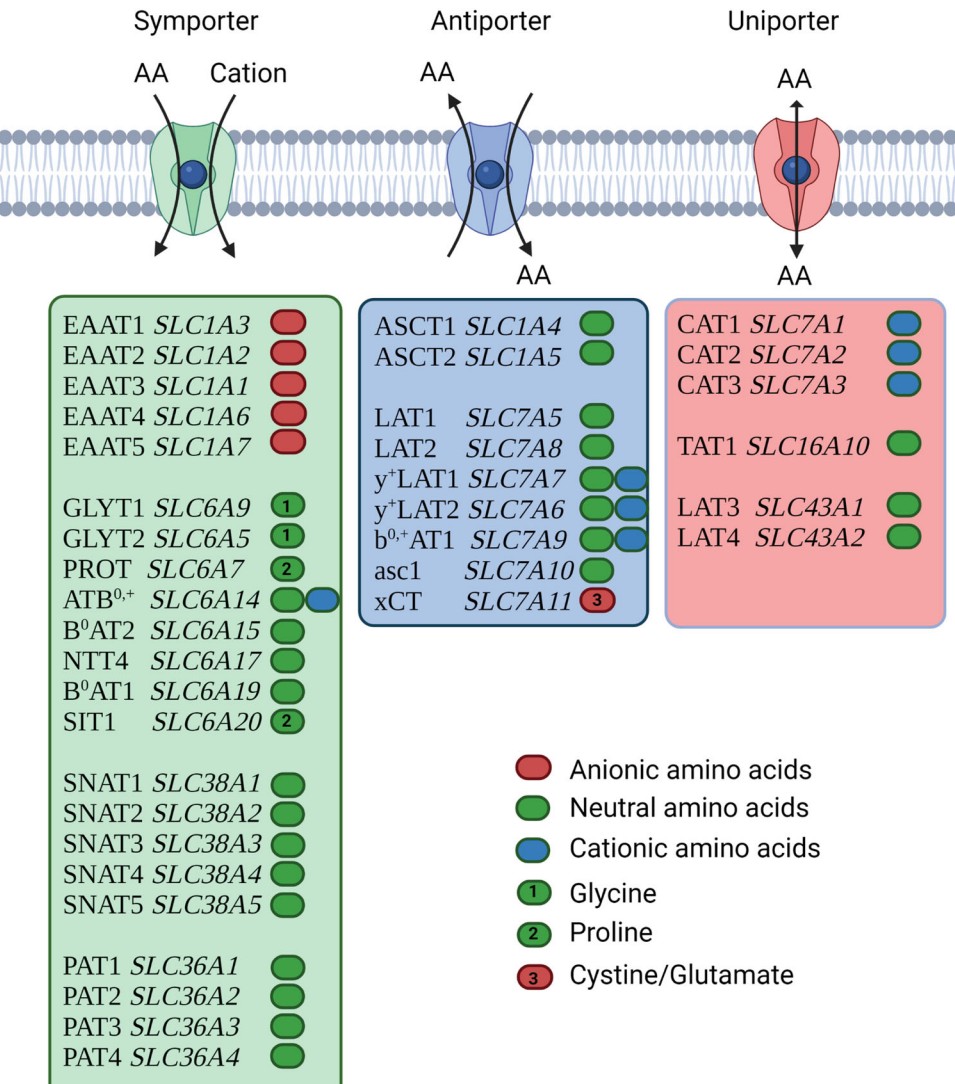

**Fig. 1 Overview of plasma membrane amino acid transporters.** Symporters are shown in green, antiporters in blue, and uniporters in red. Common names and solute carrier number are listed for each transporter. The substrate specificity of each transporters is indicated by color code (green = neutral, red = anionic, and blue = cationic). Specific amino acids are labeled as indicated.

A complete analysis of amino acid transport activities in any cell type has been difficult to achieve due to overlapping substrate specificities, different affinities, and unclear subcellular localization. It is also unclear how these transporters work together to provide cells with a harmonized mix of amino acids to sustain protein translation and other critical amino acid-dependent functions.

The lack of our understanding of the relationship between media and cytosolic amino acid concentrations was highlighted in a recent review by Wolfson and Sabatini[17]. Here, we show that this relationship is mainly determined by a combination of transport processes, which generate a stable equilibrium. The exceptions in many cell lines are glutamate and aspartate, for which transport is typically slow, while metabolism is fast. The amino acid transporter combination of a cell can be deconvoluted using flux experiments and stable isotope labeling to derive kinetic parameters. These data can then be used as inputs for in silico simulation of amino acid transport, thereby providing a systems level understanding of amino acid homeostasis.

## Results

**Amino acid transporter composition in two cell lines**. We chose two human cancer cell lines, namely, A549 lung cancer cells and U87-MG glioma cells as models for this study, because they are widely used and have different tissue origins. To qualitatively survey the amino acid transporter set in each cell line and to identify candidates for transport analysis, we analyzed public databases and used RT-PCR to determine the expression of amino acid transporters at the transcriptomic level (Figs. S1 and S2). For each transporter, a qualitative expression score was generated. In the second step, we used antibodies combined with surface biotinylation to identify transporters that are potentially active (Figs. S1, S3, and S4). Surface biotinylation/western blotting was an important verification for two reasons. Firstly, mRNA levels did not always correlate with surface expression. For instance, SNAT2 (Na$^+$-AA symporter for polar neutral AA) mRNA is abundant, but its protein can barely be detected at the cell surface in fresh media, as shown previously in 143B cells[18]. This was also observed for A549 and U87-MG cells (Figs. S2–S4). SNAT5 (SLC38A5, Na$^+$-AA symporter/H$^+$antiporter for polar neutral AA), by contrast, has a strong signal in western blots, but its mRNA was barely detectable. Secondly, some transporters are predominantly lysosomal, such as PAT1 (SLC36A1, proton AA symporter for small neutral AA), but may also occur at the cell surface in some cell types[19]. Transporters strongly expressed at the cell surface in both cell lines were: ASCT2 (antiporter for polar neutral AA), 4F2hc (SLC3A2, transporter-associated trafficking subunit) and its associated transporters LAT1 (antiporter for large neutral AA), y$^+$LAT2 (antiporter for neutral (+Na$^+$) and cationic AA), xCT (glutamate–cystine antiporter), and symporters SNAT1 (Na$^+$-AA symporter for polar neutral AA) and SNAT5. In addition, there were cell-specific transporters, such as B$^0$AT2 (SLC6A15, Na$^+$-AA symporter for neutral AA) and ASCT1 (antiporter for small polar neutral AA) in U87-MG cells. The initial survey simplifies the design and interpretation of the radioactive flux experiments outlined in the next paragraph.

**Amino acid transport activity in U87-MG cells**. After cataloging transporter candidates, amino acid transporter activity was quantified in this study using transport experiments, with radiolabeled amino acids. Table S1 lists all transport conditions to identify specific transporters. Amino acid transporters typically accept groups of related amino acids, such as large nonpolar neutral, small neutral, cationic, and anionic[20]. To capture all amino acid transporters in a cell, we used substrates to cover these groups, namely,

leucine, glutamine, alanine, arginine, glycine, proline, and glutamate. To discriminate between individual transporters, ion dependence, inhibitors and amino acid competition assays were used. Ion dependence was analyzed by replacing Na$^+$ with the impermeable organic cation N-methyl-D-glucamine$^+$ (NMDG) or with Li$^+$, immediately before addition of radiolabeled substrate. A breakdown of leucine transport in U87-MG cells is shown in Fig. 2a. In the inhibitor experiments, blocked transporters are shown above the bar, active transporters within the bar. Transporters are color coded. White gaps indicate potentially unresolved transport activity or incomplete inhibition, but these were typically small or associated with certain inhibitors, such as 2-amino-2-norbornane-carboxylic acid (BCH). Leucine transport was largely mediated by the Na$^+$-independent transporter LAT1, as evidenced through its inhibition by the specific inhibitor JPH203[21]. The Na$^+$-dependent uptake of leucine in U87-MG cells was mediated by the branched-chain amino acid transporter B$^0$AT2 and by the exchanger y$^+$LAT2—the only Na$^+$-dependent BCAA transporters expressed in U87-MG cells. Their activity can be deduced from the fraction inhibited by loratadine (B$^0$AT2)[22] and arginine (y$^+$LAT2). Accordingly, leucine uptake was completely abolished in the presence of JPH203 in Na$^+$-free transport buffer. SNAT2 can also mediate leucine transport, but its contribution as assessed by the amino acid analog N-methyl-aminoisobutyric acid (MeAIB)[23] was too small to be significant. MeAIB is a well-known inhibitor of SNAT1/2. The nonspecific amino acid analog BCH inhibits LAT1/2 and B$^0$AT2, thereby explaining its strong effect on leucine uptake. In combination, this accounted for all of leucine uptake. We did not analyze aromatic amino acid transport separately, because of the complete overlap with leucine accepting transporters in U87-MG cells.

In contrast to leucine transport, uptake of glutamine was largely mediated by Na$^+$-dependent transporters (Fig. 2b). Glutamine is a poor substrate of LAT1 and accordingly, inhibition by JPH203 was small. A small component of Na$^+$-independent glutamine uptake was allocated to LAT2 (SLC7A8). A fraction of glutamine uptake was inhibited by MeAIB, indicating that it was facilitated by either SNAT1 or SNAT2. Betaine, which discriminates between SNAT1 and SNAT2, had no effect, suggesting SNAT1 to be the dominant MeAIB-sensitive glutamine transporter. This is also in agreement with the lack of MeAIB inhibition on leucine transport. Loratadine was used to identify the fraction of glutamine uptake mediated by B$^0$AT2[22]. A particular feature of glutamine transporters SNAT3 (SLC38A3) and SNAT5 is their ability to retain transport activity, when NaCl is replaced by LiCl[11]. This was supported by glutamine transport in LiCl being higher than the glutamine transport in the absence of Na$^+$. In the absence of SNAT3 expression (Fig. S1), this component was attributed to SNAT5. All of glutamine transport was sensitive to inhibition by alanine, suggesting that the remaining Na$^+$-dependent uptake was mediated by ASCT2, a dominant neutral amino acid transporter in all cancer cells[13]. Despite significant efforts, reliable specific inhibitors of ASCT2 are yet to be developed[24], but the remaining transport activity was consistent across all conditions. These contributions account for all of glutamine uptake, and except for SNAT5, concur with mRNA expression data.

Alanine uptake (Fig. 2c) was also largely Na$^+$-dependent and mediated by a similar range of transporters as glutamine uptake. It was partially sensitive to inhibition by MeAIB (SNAT1/2), arginine (SNAT4 (SLC38A4) and y$^+$LAT2), and loratadine (B$^0$AT2), but not by JPH203 (LAT1) consistent with the substrate specificity of LAT1. Thus, the small uptake activity observed in the absence of Na$^+$ was assigned to LAT2. The related transporter ASCT1 excludes glutamine and as a result, we assigned the glutamine-resistant portion of alanine transport to ASCT1. A small lithium-dependent component was attributed to SNAT5. The large remaining fraction

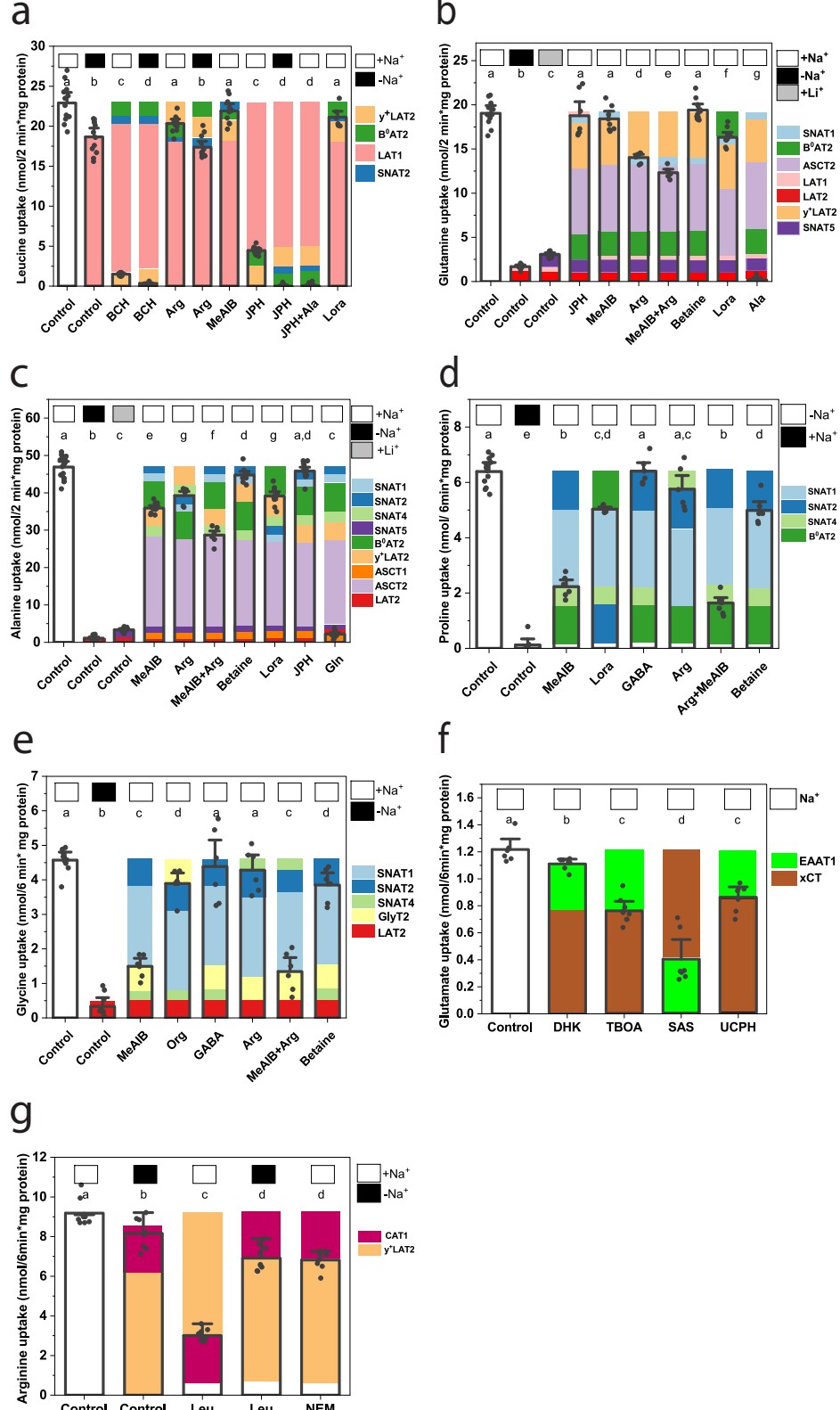

was allocated to ASCT2. These contributions accounted for all of alanine uptake.

Proline (Fig. 2d) and glycine (Fig. 2e) often have specific transporters, because they tend to be poor substrates of other amino acid transporters. Proline transport was entirely Na$^+$-dependent, excluding proton-dependent transporters of the SLC36 family (PAT1–4, SLC36A1–4). This simplifies the

interpretation of MeAIB inhibition, which affects SNAT1/2 and PAT1/2. Inhibition by betaine was less strong than by MeAIB, suggesting that SNAT1 was the dominant proline transporter. We attributed the arginine-sensitive portion of proline uptake to SNAT4. Moreover, the combination of MeAIB and arginine was more powerful than MeAIB alone. GABA did not inhibit proline transport, confirming the absence of PAT1. The fraction of

**Fig. 2 Discrimination of amino acid transport activities in U87-MG glioma cells.** The transport of [$^{14}$C]amino acids is shown for **a** leucine (100 µM, 2 min, $n$ [by column] = 12,12,12, 6, 12, 12, 12, 10, 12, 12, 15, 6; $e = 3$), **b** glutamine (100 µM, 2 min, $n$ = 12, 7, 9, 7, 8, 6, 6, 12, 11, 6; $e = 3$), **c** alanine (300 µM, 2 min, $n$ = 18, 9, 9, 10, 6, 6, 9, 11, 13, 15; $e = 3$), **d** proline (100 µM, 6 min, $n$ = 12, 6, 6, 6, 6, 6, 9, 6, 9; $e = 3$), **e** glycine (100 µM, 6 min, $n$ = 10, 9, 6, 6, 6, 6, 6; $e = 3$), **f** glutamate (100 µM, 6 min, $n$ = 6, 6, 9, 7, 7; $e = 3$), and **g** arginine (100 µM, 6 min, $n$ = 9, 8, 9, 9, 8; $e = 3$). AA uptake was measured in Na$^+$-based buffer (white squares above the bars), NMDG$^+$-based buffer (black squares above the bars), or Li$^+$-based buffer (gray squares above the bars). In the control experiments, no inhibitor or competitor was added. Except for NEM (10 min preincubation), inhibitors and competitors were added together with the substrate at the concentrations given in Table S1. To facilitate analysis, blocked transporter(s) relevant to the cell line are shown above the bars, the active transporter(s) are shown inside the bar. The transporters are color coded as indicated in the figure legend, white gaps indicate an unresolved discrepancy. Nonspecific uptake/binding was evaluated by addition of 10 mM unlabeled substrate and was subtracted from all experiments. The mean and 95% confidence intervals are shown for all data. A one-way ANOVA with Tukey's multiple comparison test was used to analyze differences between groups. Groups that were not considered different from each other were assigned the same letter, $p$ values for all comparisons are listed in Table S15. Amino acids are shown in three-letter code; BCH 2-amino-2-norbornane-carboxylic acid, DHK dihydrokainate; JPH JPH203, Lora loratadine, MeAIB *N*-methyl-aminoisobutyric acid, NEM *N*-ethylmaleimide, Org ORG25543, SAS sulfasalazine, TBOA DL-*threo*-β-benzyloxyaspartate, UCPH UCPH-101.

proline transport inhibited by loratadine was attributed to B$^0$AT2. Together these transporters explained proline uptake in U87-MG cells. Because the selectivity of loratadine has not been extensively tested against other transporters, we confirmed the contribution of B$^0$AT2 to proline, alanine, and glutamine transport by RNAi-mediated silencing, which matched the effect of loratadine (Fig. S5). Glycine transport was also dominated by SNAT1 and SNAT2, but glycine showed residual transport in the absence of Na$^+$, which was assigned to LAT2. Inhibition by arginine was not significant suggesting that SNAT4 did not contribute to glycine uptake. A small fraction of glycine transport was inhibited by ORG24453 (ref. [25]), which was consistent with a weak protein expression of GlyT2 (SLC6A5; Fig. S3).

Glutamate uptake (Fig. 2f) was mediated by xCT, which is sensitive to inhibition by sulfasalazine[26], and to a smaller extent by EAAT1, which is sensitive to inhibition by UCPH-101[27]. The inhibition by UCPH-101 matched the fraction of transport blocked by the nonspecific excitatory amino acid transporter (EAAT) inhibitor DL-*threo*-β-benzyloxyaspartate[28], suggesting that no other EAAT was involved.

Arginine uptake (Fig. 2g) was entirely Na$^+$-independent. A significant fraction of arginine transport was inhibited by leucine in the presence of Na$^+$, but not in its absence, a hallmark of y$^+$LAT2. This transporter is resistant to treatment with *N*-ethylmaleimide (NEM), while the canonical cationic amino acid transporters CAT1/2 are sensitive. Due to the relative abundance of its mRNA, the NEM-sensitive portion was assigned to CAT1. The combination of these transporters fully accounted for arginine uptake.

**Amino acid transporter activity in A549 cells.** Expression of mRNA and protein suggested a slightly simpler array of amino acid transporters in A549 cells, including ASCT2, LAT1/2, y$^+$LAT2, CAT1, xCT, and SNAT1/2/5 (Fig. S1). We used the same strategy to analyze amino acid uptake in the human lung cancer cell line A549 (Fig. 3). Leucine transport (Fig. 3a) was mediated by LAT1, LAT2, y$^+$LAT2, and SNAT2. LAT1, LAT2, ASCT2, and SNAT1/5 were the main glutamine transporters (Fig. 3b) and alanine transporters (Fig. 3c). Proline uptake was mediated by SNAT1/2/4 and ASCT1 (Fig. 3d), while glycine uptake was mediated by SNAT1/2 and ASCT2 (Fig. 3e). Glutamate transport was carried entirely by xCT (Fig. 3f). Arginine uptake was mediated by y$^+$LAT2 and CAT1 (Fig. 3g).

**Overview of transporter activity.** Figure 4 shows a summary of allocations of each transporter to amino acid fluxes in U87-MG cells (Fig. 4a, b) and in A549 cells (Fig. 4c, d). The data confirm the dominance of the antiporters LAT1 for leucine, ASCT2 for glutamine and alanine, and xCT for glutamate. Notably, uptake via xCT is nonproductive glutamate/glutamate exchange, because

cystine is reduced to cysteine in the cytosol, leaving only glutamate as an exchange substrate.

In both cell lines, transport of glutamine, alanine, and leucine was much faster than that of arginine, proline, and glutamate when measured at a concentration of 100 µM (Fig. 4e, f). Aspartate transport was barely detectable, confirming very low levels of EAATs. This dataset provided the basis of a systems level simulation of cellular amino acid transport to understand their role in amino acid homeostasis.

**Computational simulation of cellular amino acid transport.** Amino acid transport can be described by Michaelis–Menten kinetics after incorporating electrochemical driving forces. As a result, transporter simulations need to follow kinetic and thermodynamic principles. The transport mechanism for each known amino acid transporter is shown in Table S2. A curated list of $K_M$-values of human amino acid transporters is shown in Table S3. These $K_M$-values and the $V_{max}$-values derived from the transport analysis were used to simulate amino acid transport in mammalian cells using a number of established principles:

(1) Saturation of the transporter by each of its substrate amino acids (AA$_i$) follows a simple binding algorithm:

$$\text{Saturation} = [\text{AA}_i]/(K_M + [\text{AA}_i]) \qquad (1)$$

(2) Competition between substrates is incorporated by calculation of an apparent $K_M$ for amino acid ($i$) competing with other amino acid ($a$) substrates of the transporter:

$$K_{apps,i} = K_{Mi}\left(1 + \sum \frac{\text{AA}_a}{K_{Ma}}\right) \qquad (2)$$

(3) A fractional saturation is calculated for each amino acid (AA$_i$) using its apparent $K_M$.

(4) Saturation of the transporter by cotransported ions follows the Hill-equation:

$$\text{Saturation} = ([\text{Ion}_i]^n/(K_M^n + [\text{Ion}_i]^n) \qquad (3)$$

(5) Translocation of a charged complex is affected by the membrane potential[29]. The transport rate was multiplied by

$$\beta = e^{\left(-\frac{zF0.5\triangle\psi}{RT}\right)} \qquad (4)$$

when translocation is favored by the membrane potential and divided by $\beta$, when translocation occurs against the membrane potential.

Using these principles, we devised a set of equations for any kind of transport process (List S4). A couple of conditions apply to input parameters to avoid violating the second law of thermodynamics in the simulations.

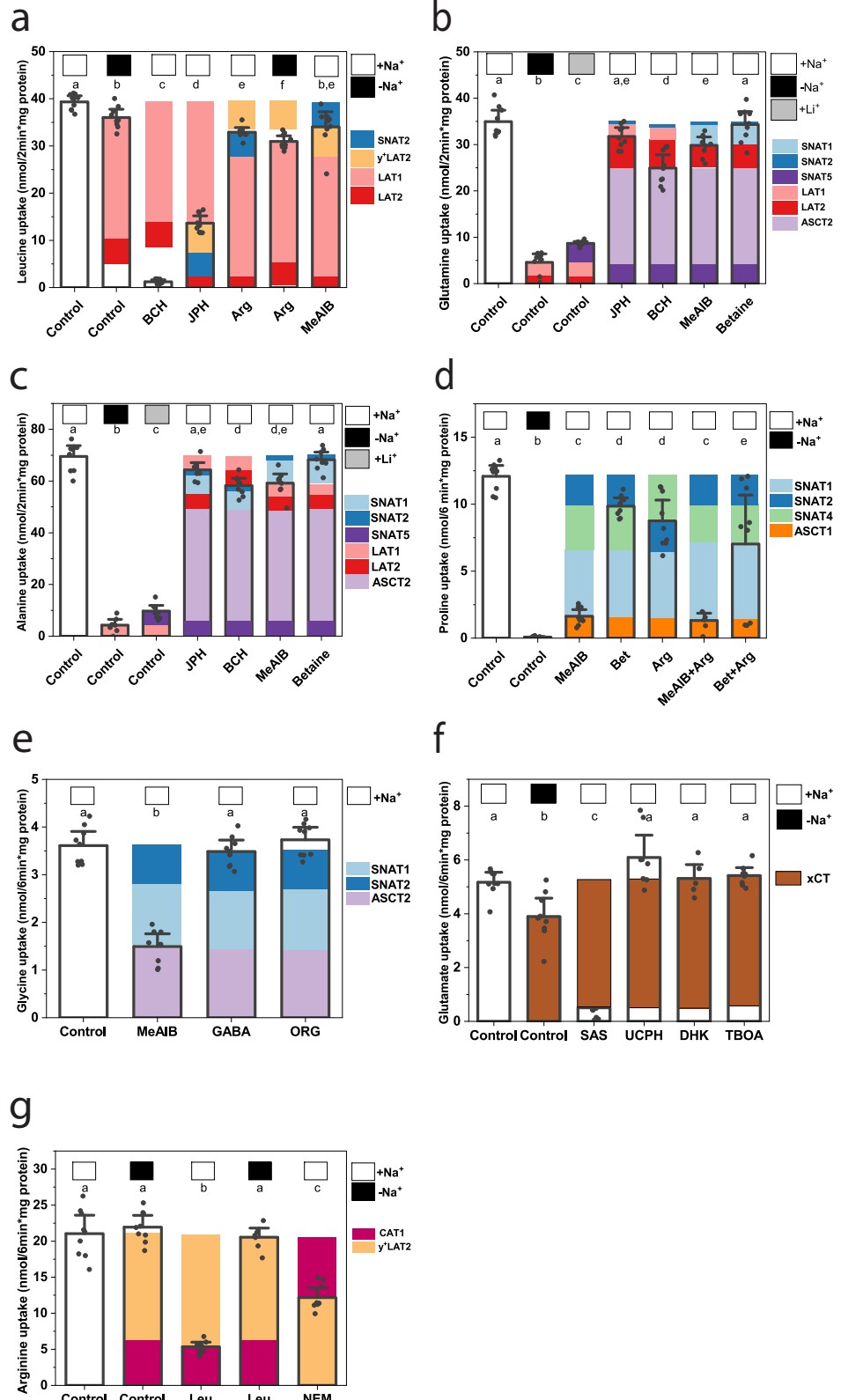

(1) All transporters: the same calculated $V_{max}$ is applied to the forward and backward flux.

(2) Uniporter: the extracellular and intracellular $K_M$-values must be equal to avoid accumulation without a substrate gradient.

(3) Symporter: the extracellular and intracellular $K_M$-values must be equal to avoid accumulation of substrate when the

membrane potential is 0 and no substrate gradient is applied. Moreover, $\beta$ (and $1/\beta$) must decrease as the substrate concentration rises on the opposite side, because the transporter will increasingly operate in electroneutral exchange mode. This is achieved by multiplication of $\beta$ (or $1/\beta$) with the fraction of transporters that can carry out net

**Fig. 3 Discrimination of amino acid transport activities in A549 lung carcinoma cells.** The transport of [$^{14}$C]amino acids is shown for **a** leucine (100 μM, 2 min, $n = 9$), **b** glutamine (100 μM, 2 min, $n = 9$), **c** alanine (300 μM, 2 min, $n = 9$), **d** proline (100 μM, 6 min, $n = 9$), **e** glycine (100 μM, 6 min, $n = 9$), **f** glutamate (100 μM, 6 min, $n = 9$), and **g** arginine (100 μM, 6 min, $n = 9$). AA uptake was measured in Na$^+$-based buffer (white squares above the bars), NMDG$^+$-based buffer (black squares above the bars), or Li$^+$-based buffer (gray squares above the bars). In the control experiments, no inhibitor or competitor was added. Except for NEM (10 min preincubation), inhibitors and competitors were added together with the substrate at the concentrations given in Table S1. To facilitate analysis, blocked transporters relevant to the cell line are shown above the bars, the active transporters are shown inside the bar. The transporters are color coded, white gaps indicate an unresolved discrepancy. Nonspecific uptake/binding was evaluated by addition of 10 mM unlabeled substrate and was subtracted from all experiments. The mean and 95% confidence intervals are shown for all data. All samples were derived from $e = 3$ experiments. A one-way ANOVA with Tukey's multiple comparison test was used to analyze differences between groups. Groups that were not considered different from each other were assigned the same letter, $p$ values for all comparisons are listed in Table S15. Amino acids are shown in three-letter code; BCH 2-amino-2-norbornane-carboxylic acid, DHK dihydrokainate, JPH JPH203, Lora, loratadine, MeAIB $N$-methyl-aminoisobutyric acid, NEM $N$-ethylmaleimide, Org ORG25543, SAS sulfasalazine, TBOA DL-*threo*-β-benzyloxyaspartate, UCPH UCPH-101.

transport. Any symporter must not exceed thermodynamic limits of accumulation as demonstrated in Table S5.

With these rules applied, we devised a computational model, which was implemented in MATLAB (called JDFC from hereon) and uses the following inputs:

(1) $K_M$-values for each substrate of each expressed transporter (Table S3). Missing $K_M$-values were estimated (est).

(2) Calculated $V_{max}$ values derived from the experiments outlined above (Table S6).

(3) Extracellular amino acid concentrations were fixed according to the media formulation.

(4) Initial intracellular amino acid concentrations were variables, but were based on actual measurements. The equilibrium is independent of the initial concentrations.

(5) Cell volume as determined by Coulter counter analysis (Fig. S6). No correction for organellar space was made, which is <10% v/v in A549 cells[30].

(6) Optional: fractional conversion of one amino acid into another (e.g., glutamine into glutamate).

(7) Optional: fractional depletion of an amino acid (metabolism or protein synthesis).

(8) A time step used for looping the algorithm.

(9) The total number of iterations.

JDFC employs a set of functions for each type of transporter (Table S2/List S4). After compiling all functions for a set of transporters in a cell line, JDFC calculates the fractional saturation of each substrate for each transporter. The corresponding portion of the total flux is allocated to each amino acid. For each substrate amino acid, a small flux was calculated for each participating transporter per time step. These values are then divided by the cell volume to generate incremental changes of AA concentrations, which are summed up for all amino acids. The workflow of the program is shown in Fig. 5a. Incorporation of amino acids into protein was much smaller than active transport. Leucine, the most frequently encoded amino acid, was incorporated at a rate of 0–0.4 nmol/mg protein in 2 min in BME medium in A549 cells compared to a transport rate of 40 nmol in 2 min (Fig. S7a), while its incorporation in U87-MG cell protein was below significance (Fig. S7b). As a result, we ignored this negligible effect on amino acid homeostasis. We hypothesized that this system is inherently stable and comes to an equilibrium point, a condition for a homeostatic process. This was indeed the case, and an example is shown in Fig. S7c. Two exceptions were glutamate and aspartate in the presence of an active EAAT. These transporters can theoretically accumulate >100 M of intracellular glutamate at plasma concentrations (Table S5). To avoid osmotic stress, both cell lines had minimal or no EAAT activity (aspartate uptake in Fig. 4e, f). Instead, glutamate was generated by metabolic conversion of glutamine, which we quantified using

[$^{13}$C]glutamine. In U87-MG cells, intracellular glutamine was 90% labeled after 30 min and >95% labeled within an hour (Fig. 5b), consistent with fast transport and exchange. Cytosolic glutamate was 30% labeled within 30 min, increasing to 55% within 2 h (Fig. 5b). Thus, significant production of glutamate occurs metabolically. In agreement with a functional glutaminolysis pathway, glutamate was metabolized through part of the TCA cycle to form aspartate. Of the initial glutamine, 20% was converted into aspartate in 24 h. The maintenance of aspartate and glutamate levels by metabolic conversion was further illustrated by using CB-839[31] to inhibit glutaminase in A549 cells (Fig. 5c). This caused glutamate levels to drop by 44% and aspartate to drop by 64% within 1 h, while glutamine increased by 30%. As a result, we included conversion of glutamine to glutamate in our simulations at the rates listed in Table S6. We also assumed that cystine would quantitatively convert to two cysteine once inside the cytosol. This generates net efflux of glutamate via xCT in exchange for cystine, which was replicated by the simulation. Importantly, our simulation identified a second pathway of glutamate efflux via ASCT2. A glutamate efflux pathway other than xCT has been predicted recently based on modeling of glutamine metabolism in cancer cells[32]. While the affinity of ASCT2 for glutamate is very low, it becomes a measurable substrate at cytosolic glutamate concentrations of 8–15 mM. To corroborate this finding from our simulation, we expressed ASCT2 together with glutamate transporter EAAT1 in *Xenopus laevis* oocytes. This allowed preloading of [$^{14}$C] glutamate into oocytes[33]. Subsequently, 0.5 mM glutamine and EAAT1 inhibitor UCPH-101 was added to the transport buffer to monitor glutamate efflux via ASCT2. In agreement with the simulation, we found a small but significant release of glutamate from oocytes (Fig. 5d), which could not be mediated by EAAT1 because of UCPH-101. Both transporters were actively expressed as assessed by separate uptake experiments (Fig. S7d). The use of ASCT2 as a pathway for glutamate efflux was confirmed in cell culture (Fig. 5e). To discriminate possible efflux pathways, we used the xCT inhibitor erastin and the ASCT2 inhibitor γ-glutamyl-*p*-nitroanilide (GPNA) and measured the appearance of [$^{13}$C]glutamate derived from [$^{13}$C]glutamine in the supernatant. Most of the efflux was suppressed by inhibition of xCT, and a smaller fraction by GPNA in U87-MG (Fig. 5e) and A549 (Fig. S7e) cells. Combined inhibition suppressed glutamate efflux almost completely.

**Comparison of in silico with in vitro data.** In view of the stable equilibrium reached by the simulation, we wondered whether this would be accurately reflected in vitro. Thus, we compared simulated and experimental intracellular amino acid concentrations in A549 (Fig. 6a) and U87-MG cells (Fig. 6b). Both datasets were in good agreement with experimental values as assessed by a

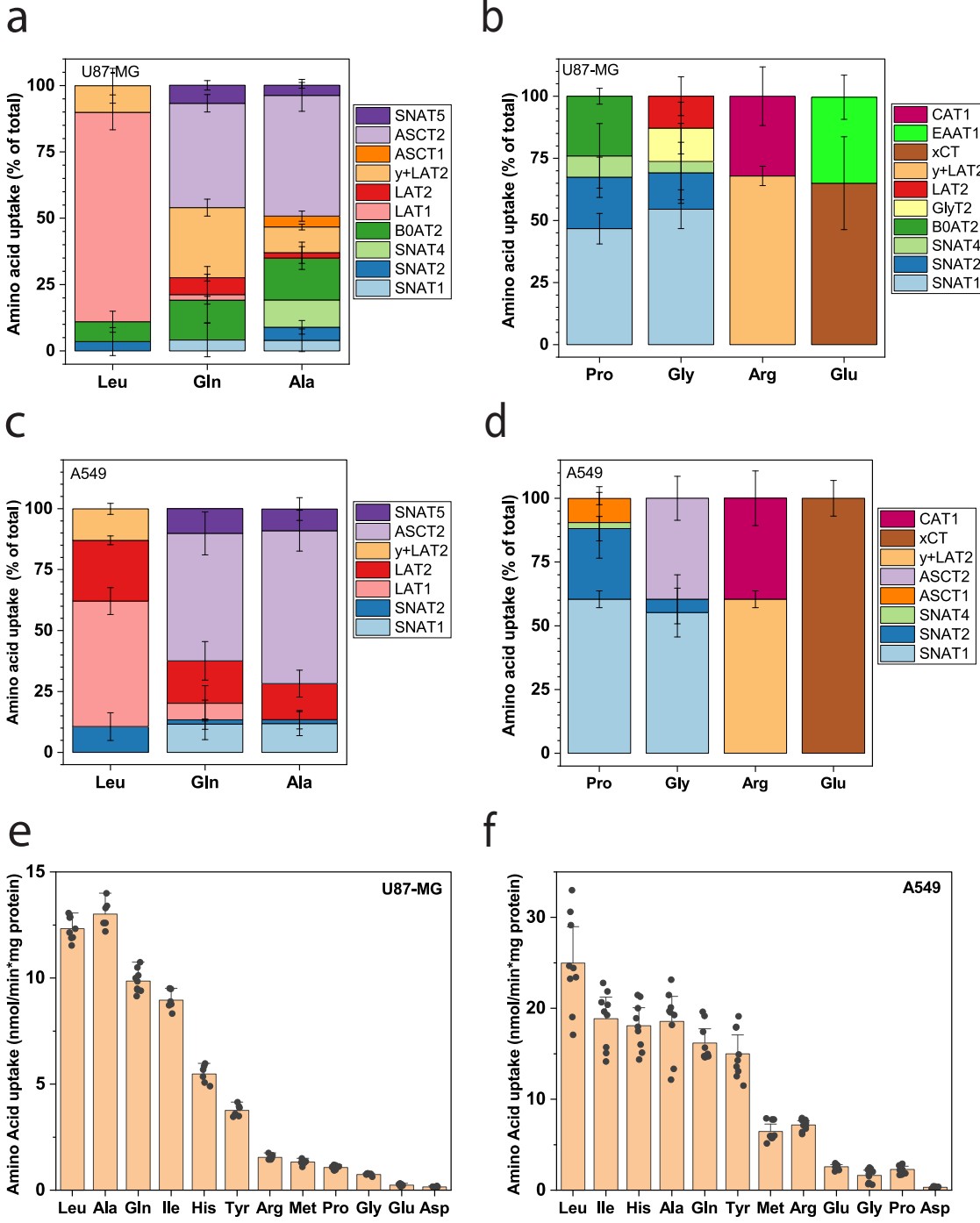

**Fig. 4 Deconvolution of plasma membrane amino acid transport.** Data from Figs. 1 and 2 were combined to illustrate the contribution of individual transporters to the uptake of the indicated amino acid. **a, b** AA transport activities in U87-MG cells are shown, **c, d** shown in A549 cells. **a, c** High-capacity transporters are shown, **b, d** low-capacity transporters are shown. Data are shown as mean ± STD, the number of biologically independent samples is the same as reported in Figs. 2 and 3. To compare the uptake activity for a variety of amino acids, uptake was measured at 100 μM in U87-MG cells (**e**, $n = 9$ biologically independent samples from $e = 3$ independent experiments) and A549 cells (**f**, $n = 9$ biologically independent samples from $e = 3$ independent experiments). 95% Confidence intervals are shown for each bar. Color coding of all transporters is indicated in the legend next to the panels.

correlation plot between experimental and predicted values, giving a Pearson's correlation coefficient of 0.9996 for A549 cells and 0.9588 for U87-MG cells (Fig. S8a, b) over a concentration range from 100 to 16,000 μM.

The endowment of cells with a mix of symporters, antiporters, and uniporters links different amino acid pools. We thus tested how amino acid concentrations would react to a fivefold spike of

one amino acid. To this end, we equilibrated U87-MG cells in a medium containing all amino acids at a concentration of 100 μM and spiked specific amino acids to 500 μM. Interestingly, the spiking affected mostly the selected amino acid (Fig. 6c). The experiment was fully replicated by the simulation (Fig. 6d), giving a Pearson's correlation coefficient ($R$) of 0.968. Glutamate levels were modulated by spiking amino acids, but there was no clear

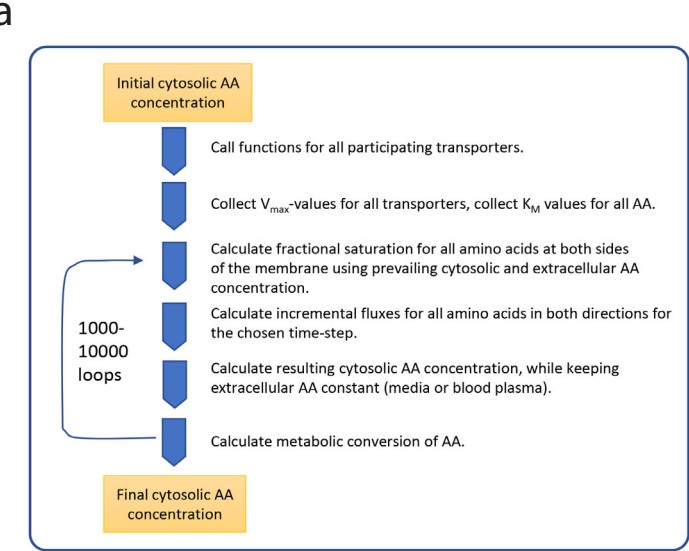

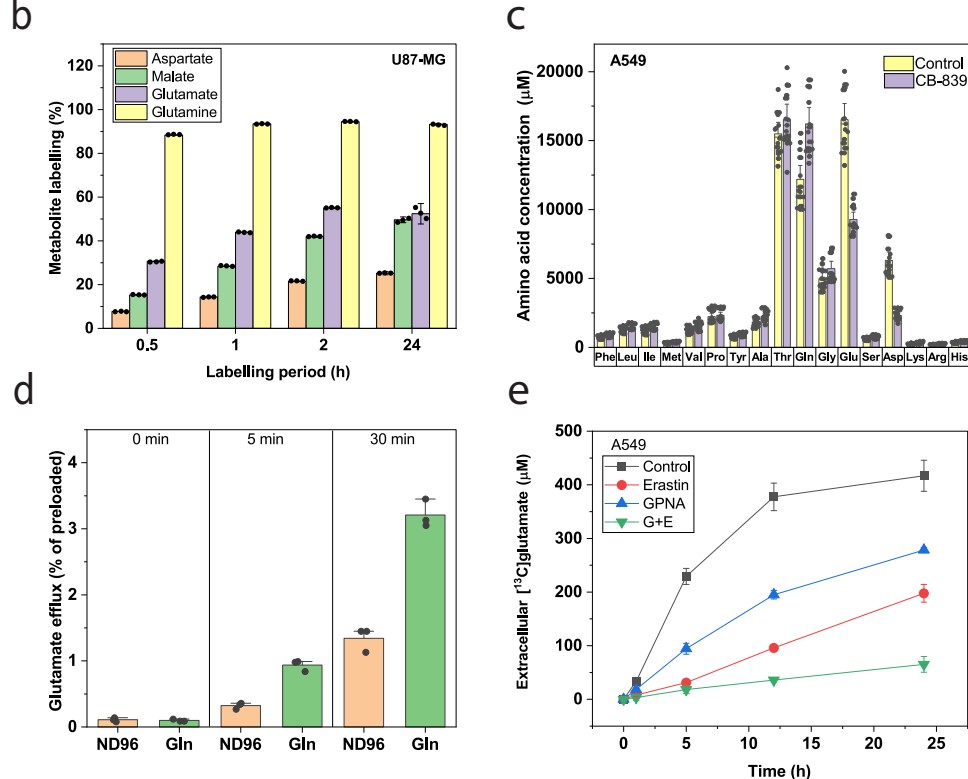

**Fig. 5 Computational simulation of amino acid transport and metabolomics. a** Principal steps of the transport simulation process. For details see text. **b** Labeling of metabolic intermediates after incubation of U87-MG cells with [$^{13}$C]glutamine (1 mM, 100% labeled). Enrichment of [$^{13}$C]metabolites is shown after 20, 60, 120 min, and 24 h ($n = 3$, $e = 3$). **c** Cytosolic amino acid concentrations were determined by LC-MS in A549 cells. Glutaminase was inhibited by incubation with CB-839 (10 μM, 1 h) and amino acid levels compared to the control ($n = 15$, $e = 3$). **d** Oocytes expressing ASCT2 and EAAT1 were preloaded with 100 μM [$^{14}$C]glutamate for 1 h. After washing, efflux was initiated by addition of 0.5 mM glutamine (green bars) to the oocyte incubation buffer. Controls remained without addition (orange bars). Recapture of glutamate was inhibited by glutamate transport inhibitor UCPH-101 ($n = 10$, $e = 3$). **e** Efflux of glutamate in A549 cells was measured by feeding cells with [$^{13}$C]glutamine in BME medium supplemented with nonessential amino acids and detecting M + 5 glutamate in the media over time ($n = 3$, $e = 3$). To discriminate between glutamate efflux pathways, xCT inhibitor erastin and ASCT2 inhibitor GPNA were used. Data are shown as mean ± 95% confidence interval.

jump of cytosolic glutamate upon increasing the extracellular concentration of glutamate itself consistent with low levels of net transport.

To further analyze the response of the simulation to extracellular amino acid changes, we added a mix of six amino acids and followed amino acid changes over 1 h in A549 cells (Fig. 6e–g). To evaluate the fit of the simulation, we used a quadrant plot of the log$_2$ (± reassigned after log conversion) of measured ($y$-axis) and simulated changes ($x$-axis). The spiked amino acids were in the upper right quadrant indicating an

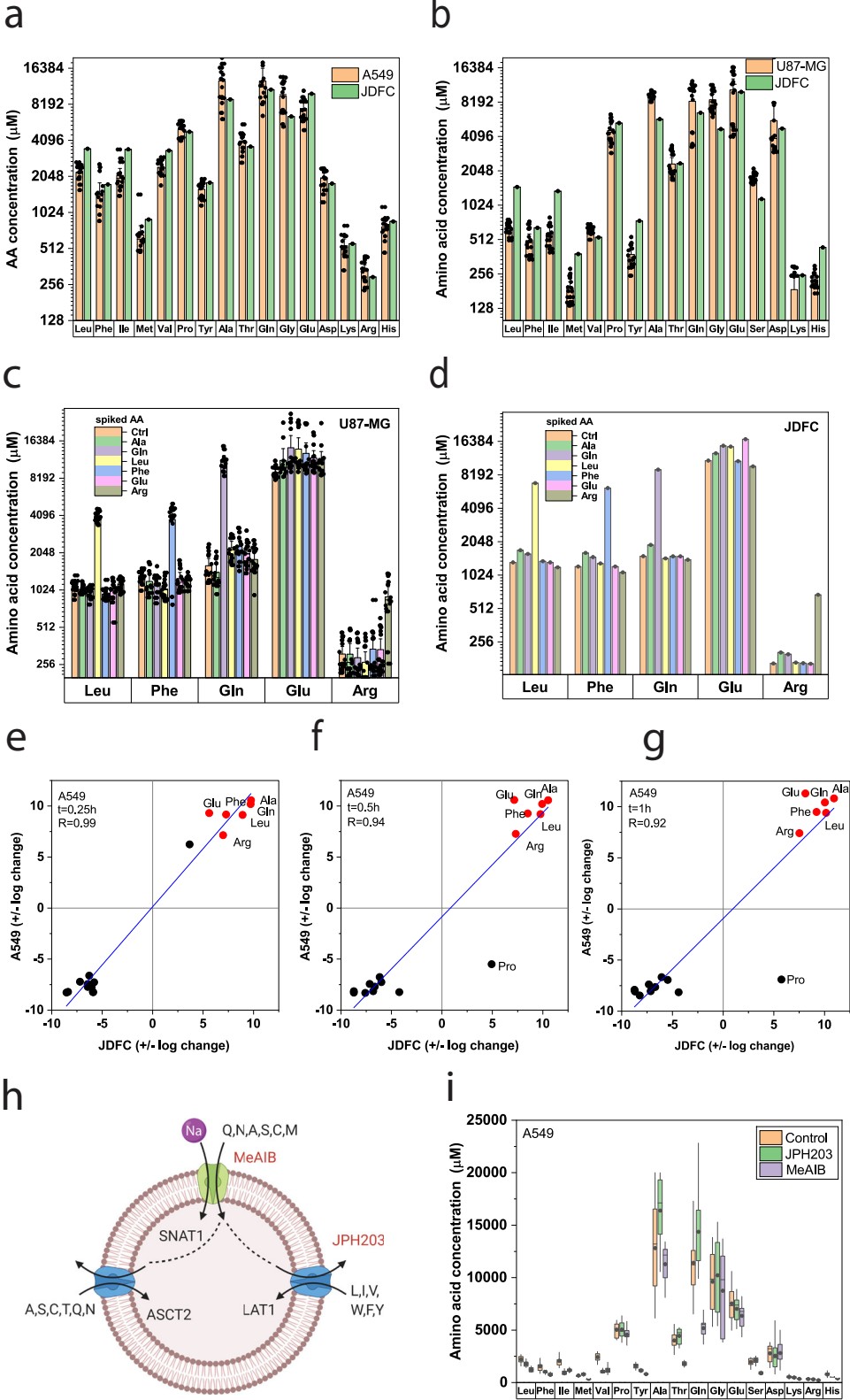

increase of the cytosolic concentrations (red dots), while non-spiked amino acids served as efflux substrates ending up in the lower left quadrant (black dots). The simulation and experimental data showed a strong correlation of $R = 0.99$, 0.94, and 0.92 at $t = 0.25$ h (Fig. 6e), 0.5 h (Fig. 6f), and 1 h (Fig. 6g), respectively. In U87-MG cells, experimental and predicted absolute amino acid concentrations showed a high level of correlation with $R = 0.95$,

0.92, and 0.89 at $t = 0.25$ h (Fig. S8c), 0.5 h (Fig. S8d), and 1h (Fig. S8e), respectively. The changes in U87-MG cells as analyzed by a quadrant plot also showed strong correlation between simulation and experiment with $R = 0.98$, 0.95, and 0.97 at $t = 0.25$ h (Fig. S8f), 0.5 (Fig. S8g), and 1h (Fig. S8h), respectively.

Our model predicts that SNAT1/2/4 accumulate amino acids in the cytosol and the imported amino acids are then used to bring

**Fig. 6 Comparison of in silico and experimental data in cell lines. a**, **b** Cytosolic amino acid concentrations were determined by LC-MS and compared to simulated (JDFC) amino acid concentrations in A549 cells (**a**, $n = 15$, $e = 3$) and U87-MG cells (**b**, $n = 15$, $e = 3$). Cells were incubated in BME medium with nonessential amino acids for 30 min before analysis. **c**, **d** U87-MG cells were incubated with a mix of all proteinogenic amino acids (100 μM each). The indicated amino acid was then spiked at a concentration of 500 μM ($n = 15$, $e = 3$). Cytosolic amino acid concentrations were determined by LC-MS in U87-MG cells (**c**) and simulated using JDFC (**d**). **e**–**g** A549 cells were preincubated for 30 min with an amino acid mixture of all proteinogenic amino acids (100 μM each). Six amino acids (red symbols) were then spiked at a concentration of 500 μM. Cytosolic amino acid concentrations were determined by LC-MS before addition of amino acid mixtures and after 15 (**e**), 30 (**f**), and 60 (**g**) min ($n = 15$, $e = 3$). The correlation between experimental data and simulation were analyzed using a quadrant chart plotting the $\log_2$ change for each time point. **h** Scheme to illustrate tertiary active transport and the action of amino acid transport inhibitors MeAIB and JPH203. Symporters are colored in green, antiporters in blue. **i** Cells were incubated for 30 min in BME plus nonessential amino acids in the absence (control) and the presence of LAT1 inhibitor JPH203 and the SNAT1/2 inhibitor MeAIB. Amino acids were measured by LC-MS after 30 min ($n = 15$, $e = 3$). Experimental data are shown as mean ± 95% confidence intervals (**a**–**c**) or as box-whisker plot (**i**) showing range, 25–75% area, median, and mean.

in other amino acids through exchange processes, i.e., tertiary active transport (Fig. 6h). Consistently, inhibition of SNAT1/2/4 by MeAIB resulted in a strong reduction of their substrates glutamine, threonine, methionine, and serine, and smaller reductions of alanine, glycine, and histidine in A549 cells (Fig. 6i). Glutamate was metabolically generated from glutamine and likewise reduced in the presence of MeAIB. Moreover, through tertiary transport, LAT1 substrates isoleucine, leucine, valine, phenylalanine, and tyrosine were reduced, as well. The increase of glutamine in the presence of JPH203 suggests that the compound blocks its efflux, and that glutamine is used as an exchange substrate in A549 cells. Inhibition of LAT1 by JPH203 (Fig. 6i) resulted in a more selective reduction of valine, tyrosine, phenylalanine, leucine, isoleucine, and histidine. A similar pattern was observed in U87-MG cells (Fig. S9a).

Amino acid mixtures are being developed as therapeutic products to rebalance metabolic dysfunction, such as muscle atrophy[34]. It is often difficult to predict cellular amino acid changes after consumption of amino acid mixtures. A complete analysis requires transport activity data, while mRNA expression data are readily available for many cell types and tissues. We thus investigated whether relative expression data were sufficient to model amino acid homeostasis in a medically relevant cell model. In this experiment, we used primary human myotubes that were incubated with an amino acid mixture containing leucine, isoleucine, valine, arginine, glutamine, histidine, lysine, phenylalanine, threonine, and N-acetyl-cysteine corresponding to AXA2678 (Axcella Health Inc.) for the improvement of muscle function[34]. Cytosolic amino acid concentrations were measured before addition of the amino acid mixture and during a time course of 120 min (Fig. 7a). Morphometry was used to determine the volume of the differentiated myotubes, and to calculate intracellular concentrations. Figure 7b shows the corresponding simulation (excluding N-acetyl-cysteine) involving 25 different amino acid transporters expressed in these cultures. For the modeling, we used transport activities proportional to mRNA expression (Table S6). The simulation correlated well with the experimental data at all time points, reaching Pearson's $R$ of 0.97 at $t = 0.25$ h (Fig. 7c) and $t = 2$ h (Fig. 7d). Significant increases were observed for all supplemented amino acids (Arg, Gln, His, Ile, Leu, Lys, Phe, Thr, and Val) within 15 min in myotubes and in the simulation (Fig. 7e, f, Pearson's $R = 0.92/0.93$), while alanine, serine, and asparagine were the major efflux substrates. The dynamics of the system were also replicated well for media inducing inflammation in muscle (sarcopenic muscle model; Fig. S9b). The myotube experiments suggest that concentration transients of cellular amino acids can be simulated based on plasma amino acid changes and mRNA expression data. Importantly, the data demonstrate that the relationship of extracellular and intracellular amino acid concentrations is determined by transport processes.

## Discussion

Transporters are typically classified as uniporters, symporters, and antiporters. Our results suggest a more functional classification based on their role in a cellular context. To understand amino acid homeostasis in mammalian cells three different types of transporters are required, namely, harmonizers, loaders, and controller transporters (Fig. 8a).

(i) Harmonizers are the main contributors to a cell's amino acid transport activity. These are rapid antiporters for a group of amino acids, such as large nonpolar amino acids (LAT1) or polar amino acids (ASCT2). To maintain a harmonized mix of all 20 proteinogenic amino acids, these transporters are significantly faster than net uptake. Consistently, LAT1 was the dominant transporter for large neutral amino acids, while ASCT2 dominated the transport of small and polar amino acids. The harmonizer y⁺LAT2 forms the nexus between neutral and cationic amino acids. In the presence of the cationic amino acid loader CAT1, y⁺LAT2 mediates efflux of cationic amino acids in exchange for neutral amino acids. Antiporters show significant asymmetry with cytosolic $K_M$-values being 100–1000-fold higher than extracellular $K_M$-values[35–37]. Moreover, certain amino acids are good uptake substrates, while their efflux is limited. Consistent with a recent report[36], we modeled a higher intracellular $K_M$ of ASCT2 for alanine, which stabilized its intracellular concentration.

(ii) Amino acid loaders accumulate amino acids in the cytosol. SNAT1, for instance, mediates the net uptake of a group of amino acids, such as small and polar neutral amino acids[11]. Strong Na⁺-dependence of alanine and glutamine uptake is an indication that their transport is mediated by loaders. Loaders must have at least one overlapping amino acid substrate with the harmonizers, but typically share several substrates. Glutamine and alanine are ideal substrates as they are highly abundant in blood plasma and are major substrates for SNAT1. SNAT1 is a Na⁺/neutral amino acid symporter using the electrical and chemical driving force of Na⁺ to accumulate its substrates in the cytosol, as shown in Eq. (5)[11]:

$$\frac{AA_{cyt}}{AA_{ext}} = \frac{Na_{ext}}{Na_{cyt}} e^{\left(-\frac{zF\triangle\psi}{RT}\right)} \tag{5}$$

Under physiological conditions, this allows for an ~100-fold accumulation inside the cytosol. In conjunction with harmonizers, SNAT1 can lift the concentration of all neutral amino acids above those observed in the blood plasma or cell culture media. Over time, SNAT1 would accumulate all neutral amino acids to the same ratio as its own substrates. Another important loader is CAT1 working in conjunction with harmonizer y⁺LAT2. Not all cells express CAT1 (ref. [10]) and in its absence y⁺LAT2 can also take up cationic amino acids as observed in some cell types[38], but the resulting cytosolic concentrations are low. This is consistent with a low ion selectivity of the transporter allowing it to use Na⁺ or K⁺ for cotransport[14].

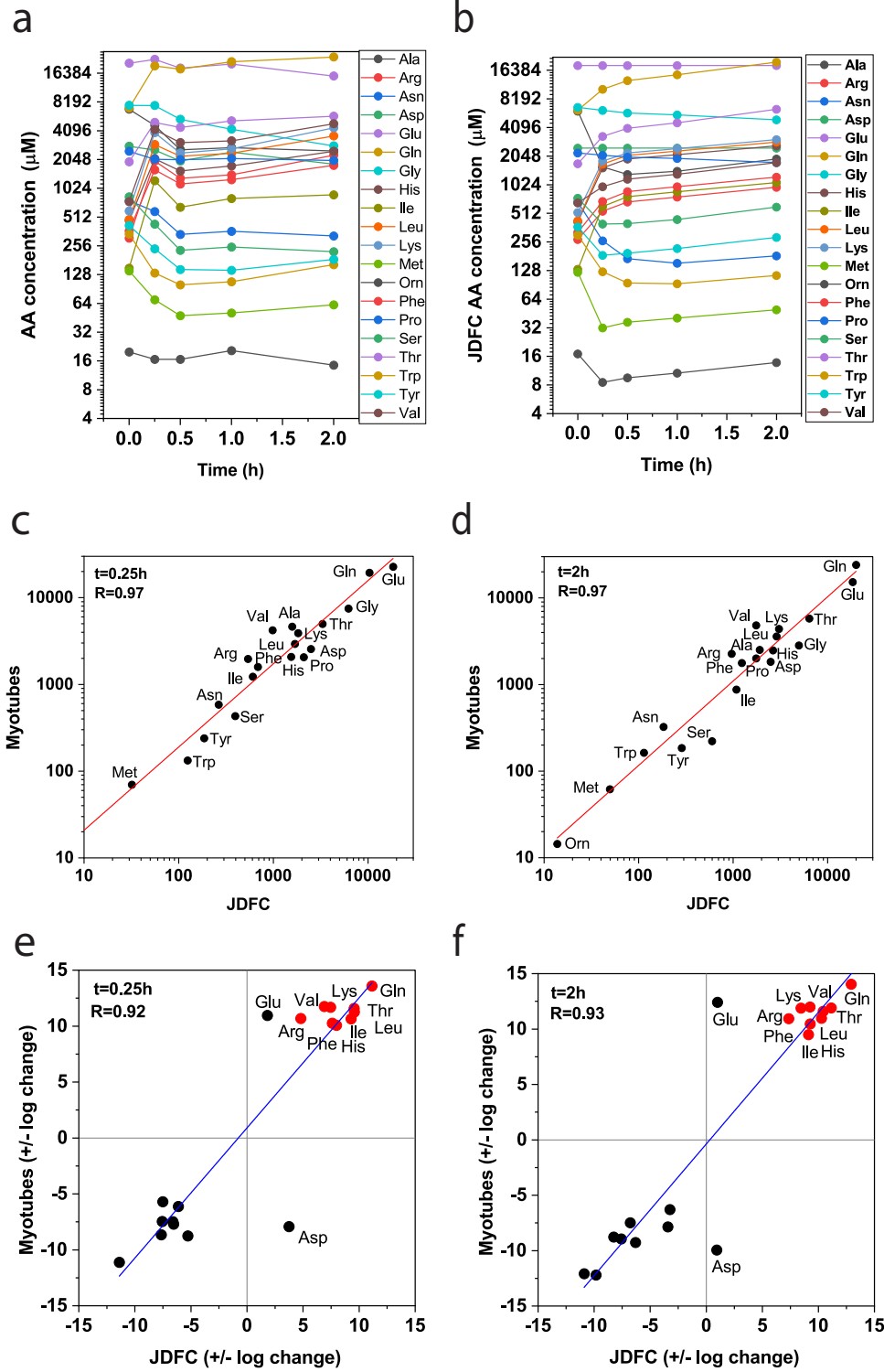

**Fig. 7 Comparison of in silico and experimental data in human myotubes.** Human primary myotubes were treated with an amino acid mixture corresponding to the constituent amino acids of AXA2678 for 2 h ($n = 3$, $e = 3$). **a** Cytosolic amino acid concentrations were determined by LC-MS before addition of amino acid mixtures and for 2 h after addition. Cell volume was determined by morphometry. The same initial concentrations were used for JDFC and the simulation was run for 3200 cycles (**b**). **c**, **d** Correlation of experimental and simulated data after $t = 0.25$ h, $R = 0.97$ (**c**) and $t = 2$ h, $R = 0.97$ (**d**). **e**, **f** Correlation analysis of changes using a quadrant chart plotting the $\log_2$ change at $t = 0.25$ h, $R = 0.92$ (**e**) and $t = 2$ h, $R = 0.93$ (**f**). Spiked amino acids are shown as red dots.

a

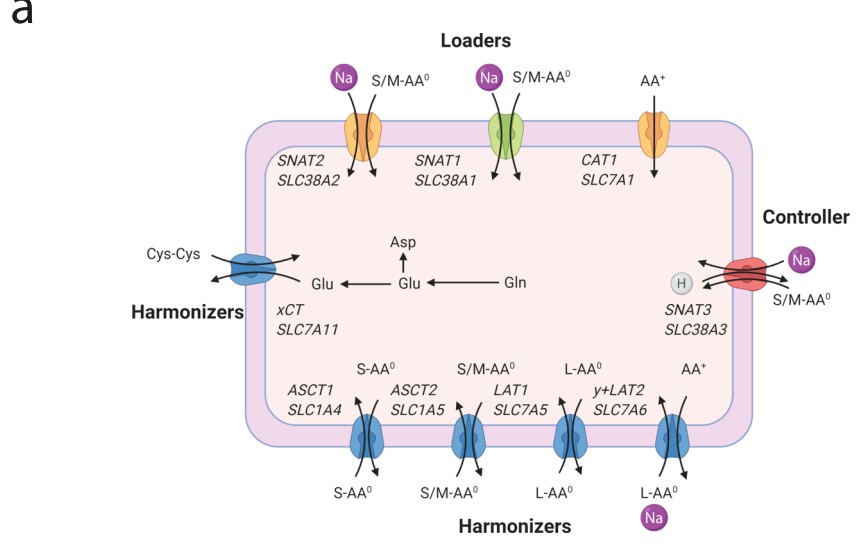

b

| A549 | ASCT2 | CAT1 | LAT1 | LAT2 | xCT | y+LAT2 | SNAT1 | SNAT2 | SNAT4 | SNAT1/ | SNAT5 |
|---|---|---|---|---|---|---|---|---|---|---|---|
| Ala | 0.37 | 0.39 | 0.99 | 1.05 | 0.98 | 0.42 | 0.94 | 0.98 | 0.91 | 0.82 | 5.53 |
| Cys | 27.84 | 0.32 | 0.97 | 0.93 | 0.54 | 0.26 | 0.92 | 0.97 | 0.88 | 0.77 | 8.12 |
| Asp | 1.00 | 1.00 | 1.00 | 1.00 | 1.00 | 1.00 | 1.00 | 1.00 | 1.00 | 1.00 | 1.00 |
| Glu | 2.11 | 0.39 | 0.98 | 0.94 | 5.40 | 0.32 | 0.94 | 0.98 | 0.90 | 0.81 | 4.46 |
| Phe | 1.26 | 0.37 | 1.05 | 0.98 | 0.98 | 0.29 | 0.95 | 0.98 | 0.92 | 0.84 | 3.83 |
| Gly | 1.66 | 0.46 | 0.98 | 0.87 | 0.98 | 0.43 | 0.94 | 0.98 | 0.86 | 0.76 | 3.03 |
| His | 1.03 | 0.40 | 1.06 | 1.00 | 0.98 | 0.39 | 0.95 | 0.98 | 0.93 | 0.84 | 3.65 |
| IsoL | 1.23 | 0.37 | 1.11 | 0.99 | 0.98 | 0.28 | 0.95 | 0.98 | 0.93 | 0.85 | 3.79 |
| Lys | 1.12 | 0.28 | 1.01 | 0.99 | 0.98 | 4.40 | 0.96 | 0.99 | 0.94 | 0.88 | 3.27 |
| Leu | 1.29 | 0.36 | 0.99 | 0.98 | 0.98 | 0.26 | 0.95 | 0.98 | 0.92 | 0.84 | 3.88 |
| Met | 1.31 | 0.37 | 0.98 | 0.98 | 0.98 | 0.27 | 0.95 | 0.98 | 0.92 | 0.84 | 3.88 |
| Asn | 1.80 | 0.42 | 0.98 | 0.91 | 0.98 | 0.38 | 0.94 | 0.98 | 0.91 | 0.82 | 3.72 |
| Orn | 1.12 | 0.27 | 1.01 | 1.00 | 0.98 | 4.27 | 0.96 | 0.99 | 0.94 | 0.88 | 3.17 |
| Pro | 0.96 | 1.23 | 1.00 | 1.00 | 1.01 | 1.24 | 0.95 | 0.92 | 0.63 | 0.32 | 0.64 |
| Gln | 1.91 | 0.41 | 0.98 | 0.94 | 0.98 | 0.33 | 0.94 | 0.98 | 0.91 | 0.83 | 3.82 |
| Arg | 1.13 | 0.29 | 1.01 | 0.99 | 0.98 | 4.44 | 0.96 | 0.99 | 0.94 | 0.88 | 3.29 |
| Ser | 1.83 | 0.42 | 0.98 | 0.93 | 0.98 | 0.38 | 0.94 | 0.98 | 0.91 | 0.82 | 3.80 |
| Thr | 8.48 | 0.41 | 0.98 | 0.88 | 0.98 | 0.36 | 0.94 | 0.98 | 0.91 | 0.82 | 4.25 |
| Val | 1.46 | 0.36 | 0.52 | 0.95 | 0.98 | 0.28 | 0.94 | 0.98 | 0.92 | 0.83 | 4.09 |
| Trp | 1.21 | 0.36 | 1.09 | 1.00 | 0.98 | 0.27 | 0.95 | 0.98 | 0.93 | 0.85 | 3.79 |
| Tyr | 1.24 | 0.37 | 1.13 | 0.99 | 0.98 | 0.28 | 0.95 | 0.98 | 0.93 | 0.85 | 3.80 |
| | H | L | H | | H | H | L | L | L | L/L/L | C |

c

| U87-MG | EAAT3 | ASCT2 | ASCT1 | B0AT2 | CAT1 | LAT1 | LAT2 | xCT | y+LAT2 | SNAT1 | SNAT2 | SNAT4 | SNAT1/2/4 | SNAT5 |
|---|---|---|---|---|---|---|---|---|---|---|---|---|---|---|
| Ala | 1.00 | 0.54 | 0.87 | 0.98 | 0.74 | 1.00 | 1.02 | 0.99 | 0.90 | 0.91 | 0.91 | 0.83 | 0.60 | 1.74 |
| Cys | 1.00 | 2.67 | 0.56 | 0.98 | 0.74 | 1.00 | 1.00 | 0.84 | 0.78 | 0.91 | 0.91 | 0.82 | 0.59 | 1.48 |
| Asp | 1.00 | 1.00 | 1.00 | 1.00 | 1.00 | 1.00 | 1.00 | 1.00 | 1.00 | 1.00 | 1.00 | 1.00 | 1.00 | 1.00 |
| Glu | 1.00 | 1.80 | 0.87 | 0.99 | 0.79 | 1.01 | 0.99 | 1.60 | 0.77 | 0.93 | 0.93 | 0.86 | 0.68 | 1.25 |
| Phe | 1.00 | 1.10 | 1.00 | 0.97 | 0.74 | 0.67 | 0.93 | 0.99 | 0.56 | 0.93 | 0.93 | 0.86 | 0.67 | 1.31 |
| Gly | 1.00 | 0.89 | 0.84 | 0.99 | 0.83 | 1.02 | 1.13 | 0.99 | 0.85 | 0.92 | 0.90 | 0.73 | 0.49 | 1.58 |
| His | 1.00 | 1.05 | 0.95 | 0.98 | 0.76 | 1.37 | 0.98 | 0.99 | 0.66 | 0.93 | 0.93 | 0.87 | 0.69 | 1.14 |
| Ile | 1.00 | 1.10 | 0.98 | 0.97 | 0.74 | 1.00 | 0.95 | 0.99 | 0.51 | 0.93 | 0.93 | 0.87 | 0.69 | 1.30 |
| Lys | 1.00 | 1.11 | 0.91 | 0.98 | 0.56 | 1.05 | 0.99 | 0.99 | 7.84 | 0.94 | 0.94 | 0.88 | 0.71 | 1.34 |
| Leu | 1.00 | 1.12 | 0.96 | 0.97 | 0.74 | 1.08 | 0.96 | 0.99 | 0.50 | 0.93 | 0.93 | 0.87 | 0.68 | 1.31 |
| Met | 1.00 | 1.12 | 0.96 | 0.98 | 0.75 | 1.11 | 0.96 | 0.99 | 0.52 | 0.92 | 0.93 | 0.86 | 0.66 | 1.30 |
| Asn | 1.00 | 2.15 | 0.81 | 0.98 | 0.77 | 1.01 | 0.98 | 0.99 | 0.78 | 0.91 | 0.91 | 0.83 | 0.60 | 1.27 |
| Orn | 1.00 | 1.11 | 0.92 | 0.98 | 0.55 | 1.05 | 0.99 | 0.99 | 7.61 | 0.94 | 0.94 | 0.88 | 0.72 | 1.33 |
| Pro | 1.00 | 0.78 | 2.20 | 0.97 | 0.82 | 0.94 | 1.00 | 0.99 | 0.91 | 0.91 | 0.87 | 0.75 | 0.46 | 1.45 |
| Gln | 1.00 | 1.85 | 0.82 | 0.98 | 0.75 | 1.02 | 0.99 | 0.99 | 0.70 | 0.91 | 0.92 | 0.83 | 0.61 | 1.32 |
| Arg | 1.00 | 1.11 | 0.91 | 0.98 | 0.57 | 1.05 | 0.99 | 0.99 | 7.91 | 0.93 | 0.93 | 0.88 | 0.71 | 1.34 |
| Ser | 1.00 | 2.28 | 0.67 | 0.98 | 0.77 | 1.00 | 0.99 | 0.99 | 0.79 | 0.91 | 0.92 | 0.83 | 0.61 | 1.34 |
| Thr | 1.00 | 2.86 | 0.68 | 0.98 | 0.76 | 1.00 | 0.98 | 0.99 | 0.78 | 0.91 | 0.92 | 0.82 | 0.60 | 1.43 |
| Val | 1.00 | 0.89 | 1.56 | 0.96 | 0.71 | 0.54 | 0.93 | 0.99 | 0.65 | 0.91 | 0.91 | 0.82 | 0.59 | 1.52 |
| Trp | 1.00 | 1.10 | 0.96 | 0.98 | 0.73 | 1.37 | 0.95 | 0.99 | 0.50 | 0.93 | 0.93 | 0.87 | 0.68 | 1.31 |
| Tyr | 1.00 | 1.10 | 0.97 | 0.98 | 0.74 | 1.41 | 0.92 | 0.99 | 0.56 | 0.93 | 0.93 | 0.86 | 0.68 | 1.31 |
| | I | H | H | L | L | H | H | H | H | L | L | L | L/L/L | C |

Loaders can be actively upregulated when amino acids are depleted. SNAT2 for instance is highly regulated at the transcriptional, translational, and protein level[39]. Accordingly, we could only detect marginal activity of SNAT2 in well-nourished cells[18]. CAT1 is also actively regulated at multiple levels[40], allowing for the active import of cationic amino acids. Due to the opposing fluxes between CAT1 and y+LAT2, cationic amino

levels in A549 and U87-MG cells were only twofold higher than plasma amino acid concentrations.

(iii) Controllers limit the accumulation of amino acids to avoid excessive amino acid accumulation by loaders. Several transporters fall into this category, such as SNAT3/5 and LAT3/4. For instance, SNAT3/5 mediate Na+-amino acid symport/H+-antiport. The transport process is therefore electroneutral, and

**Fig. 8 Common amino acid transporters in mammalian cells and their role in amino acid homeostasis. a** Types of amino acid transporters in mammalian cells. Transporters are colored by function. Loaders that mediate net uptake into cells are shown in green and orange (inducible during amino acid starvation for rescue purposes). Harmonizers that exchange amino acids are shown in blue. Controller transporters that protect against excessive accumulation are shown in red. The common transporter names and SLC acronyms are shown in italics. Amino acid (AA) charge is indicated as $AA^0$, $AA^-$, or $AA^+$. Amino acid size is indicated as small (S), medium (M), or large (L). The most common transport reaction is indicated by the arrows, but all processes are reversible. Dominant metabolic conversions are shown, the three-letter code is used for individual amino acids. Cys–Cys is cystine. **b, c** Equilibrium conditions were simulated for A549 cells (**b**) and U87-MG cells (**c**) and normalized to one. Subsequently, the simulation was repeated and one or several transporters removed as listed above the table. The change of the resulting final amino acid concentration was plotted for each amino acid and displayed as a heat map with red indicating an increase of cytosolic amino acid concentration and blue indicating a decrease. H harmonizer, L loader, C controller, I inactive transporter.

accumulation of amino acids only depends on the prevailing $Na^+$ and $H^+$ gradients[41] allowing for a modest 15-fold accumulation as shown in Eq. (6).

$$\frac{AA_{cyt}}{AA_{ext}} = \frac{Na_{ext}}{Na_{cyt}} \frac{H_{cyt}}{H_{ext}} \qquad (6)$$

This is significantly less than the 100-fold accumulation generated by SNAT1. Accordingly, modeling shows that SNAT5 is a significant efflux pathway for highly abundant amino acids, such as alanine, glycine, and glutamine. Notably, the substrate affinity of these transporters is very low. As a result, efflux only becomes considerable when intracellular substrates concentrations reach several millimolar. Additional mechanisms to avoid excessive accumulation of amino acids are metabolism and trans-inhibition of amino acid loaders, such as SNAT1/2 (ref. [42]).

The classification was supported by sensitivity analysis (Fig. 8b, c). In the analysis, we removed individual transporters from the simulation to observe the effect on the amino acid pool. Removal of a loader should reduce amino acid levels (shift from red to blue), while removal of a controller should increase amino acid levels (shift from blue to red). Harmonizers generate a mixed picture or change very little because the mix is already harmonized. Consistently, loss of CAT1 and SNAT1/2/4 generated a blue shift. The redundancy between SNAT1/2/4 was clearly visible in comparison to the combined removal of all three transporters. The harmonizer y+LAT2 can raise the concentrations of a variety of neutral amino acids at the expense of cationic amino acids loaded by CAT1. CAT1 appears to be the dominant loader in A549 cells (Fig. 8b), while the load was more evenly spread in U87-MG cells (Fig. 8c). SNAT5 increases amino acid levels when removed, consistent with its controller function.

The general view that most amino acid transporters have indeed been identified, was supported in this study by a complete deconvolution of amino acid transport in two cell lines of very different origin, namely, lung and brain. One difficulty in the deconvolution process was the presence of high-capacity substrates (leucine, alanine, and glutamine) and low-capacity substrates (glutamate, arginine, glycine, and proline). Standard deviations of individual incubations of the high-capacity substrates were as large as the total transport of some of the low-capacity substrates. As a result, minor components to the flux of high-capacity substrates may be overlooked, because of a lack of statistical significance, but can often be identified through low-capacity substrates. Moreover, transporters with high catalytic rates and with low transcript copy numbers, which are efficiently translated, may be underestimated or overlooked. Together, the set of experiments we used provides significant redundancy. In general, we found that the use of canonical high-affinity substrates for each transporter was ideal to estimate $V_{max}$.

A consistent observation was the very low activity of EAAT-type glutamate transporters. In the brain, extracellular glutamate levels are <1 μM, while cytosolic levels are 5–10 mM (ref. [43]). In this environment, glutamate transporters can work close to

equilibrium, while glutamate concentrations in blood plasma and cell culture media are ~100 μM. This poses an osmotic threat to cells in a high glutamate environment. Consequently, cancer cells express EAATs at very low levels and produce glutamate metabolically. To balance the production, glutamate is released in exchange for cystine via xCT and via ASCT2. While the intracellular glutamate $K_M$ of xCT and ASCT2 are unknown, they are likely to be very high to allow accumulation of close to 10 mM in the cytosol. This is corroborated by the simulation and by experimental observations of highly asymmetric $K_M$-values in several antiporters[35,37,44]. Cancer cells generate glutamate and aspartate via glutaminolysis[24,45,46] providing precursors for nucleobase synthesis or to synthesize asparagine, which can be used as an exchange substrate for antiporters, such as ASCT2 (ref. [47]). A limited understanding of proline efflux may underlie the poor prediction of proline dynamics in A549 cells.

Transporters primarily determine amino acid equilibrium concentrations because uptake and efflux rates are much faster than those of metabolism and protein synthesis. During transport experiments in uptake buffer, cell growth, and net amino acid incorporation into protein was negligible. If an amino acid is metabolized at a low rate inside the cell, it will be rapidly replaced by extracellular amino acids. A significant impact of metabolism was observed due to the conversion of glutamine to glutamate to maintain intracellular glutamate levels against active efflux via xCT and ASCT2. As observed in the amino acid spike experiments, transport processes will generate a mirror image of elevated plasma amino acid concentrations in the cytosol. After a meal, amino acid levels increase, and this will be translated into elevated cytosolic concentrations, which in turn activates metabolism. Over several hours this will remove amino acids thereby normalizing cytosolic and plasma amino acid concentrations. Once fasting levels are reached, metabolism slows down, particularly that of essential amino acids. At any time, transport processes hold the ratio between cytosolic and plasma amino acid concentrations constant. When there is uneven use of amino acids for protein synthesis in the cytosol the harmonizing process will reestablish concentrations that mirror blood plasma (or media). The constant relationship between extracellular and intracellular amino acid concentrations also explains why glycine is released in slowly growing cancer cell lines because there is excess glycine, while it is consumed in fast growing cell lines, when net demand produces an imbalance in the cytosol[48].

The results shown here have implications for amino acid signaling. Sestrin2 and CASTOR1/2 are considered sensors for leucine and arginine, respectively[17]. Sestrin binds to leucine with a $K_d$ of 20 μM (ref. [49]) and castor binds arginine with a $K_d$ of 30 μM (ref. [50]). As shown here, cytosolic equilibrium concentrations of leucine are several-fold higher than in blood plasma, typically reaching 1–3 mM. The mechanism by which sestrin or CASTOR modulate mTORC1 may therefore include non-agonist competition with other amino acids or other mechanisms, which modify the apparent affinity of these sensors.

The lysosomal arginine sensor SNAT9 (refs. [51–53]) has lower amino acid affinity, but amino acid levels in lysosomes are unknown[17]. It is worth noting that several amino acid transporters, such as SNAT2, SNAT9, and PAT4 are considered transceptors and shown to regulate mTORC1 independent of their transport function[54,55]. Consistently, mTORC1 is activated upon addition of amino acids before substantial changes of intracellular amino acid levels are observed[56]. Structural evidence for the transceptor concept has recently emerged[57]. Amino acid signaling interacts with the amino acid transportome. GCN2 signaling increases expression of loaders and harmonizers, such as SNAT2 (ref. [18]), xCT, ASCT1, ASCT2 (ref. [58]), and CAT1 (ref. [59]). mTORC1 also appears to increase the transcription of a subset of ATF4 activated genes, including amino acid transporters xCT, LAT1, CAT1, ASCT2, ASCT1, and SNAT1 (ref. [60]).

We have shown here that amino acid transporters play a key role in maintaining elevated pools of amino acids within the cytosol, and establish a relationship between extracellular and intracellular amino acid levels, which has been elusive heretofore[17]. The exception in the studied cell lines are glutamate and aspartate, for which transport is typically slow, while metabolism is fast. The methodology can be extended to include organellar transport processes, such as lysosomal and mitochondrial amino acid transport if volume data are available and transporters are similarly well characterized as plasma membrane transporters. Moreover, it can be used to model other types of transport in mammalian cells, such as drug transport. For the first time, this allows for a system-level understanding of amino acid homeostasis in mammalian cells that can be integrated with modeling of cellular metabolism.

## Methods

Chemicals, Media, Peptides, recombinant proteins, and assay kits are listed in Table S7. Cell lines and primary cells are listed in Table S8. Software and script are listed in Table S9.

**Cell culture**. Human lung carcinoma A549 cells were kindly gifted by Ross Hannan (John Curtin School of Medical Research, Australian National University) and human glioblastoma U87-MG cells were purchased from ATCC. A549 cells were cultured in DMEM/F12 supplemented with 10% fetal bovine serum (FBS), 2 mM glutamine, 10 U/mL penicillin, and 10 μg/mL streptomycin. U87-MG cells were cultured in supplemented BME containing 2 mM glutamine, 10% FBS, 1 mM sodium pyruvate, 10 U/mL penicillin, and 10 μg/mL streptomycin (all obtained from Gibco) and nonessential amino acids at concentrations described previously[18]. Primary human myoblasts (PHMs) from one healthy male Caucasian donor (Lonza, Donor ID HSMM, age 29, BMI 35.7) were previously screened, and validated based on-post thaw viability and myotube growth. On day 0, qualified PHMs were thawed and plated in a culture flask with skeletal muscle cell growth basal medium-2 (Lonza) fortified with SingleQuots™ supplements and growth factors (Lonza). On day 1, PHMs were collected with subculture reagents (Lonza) and seeded onto collagen I-coated 96-well plates (Corning) at a density of 3200 cells/well. On day 2, the medium was replaced with DMEM supplemented with 2% horse serum and 1% penicillin–streptomycin (all obtained from Gibco) to achieve myotube differentiation, and the cells were incubated at 37 °C for 5 days prior to experimentation. To study the effects of LIVRQNacHKFT (AXA2678 constituents) on the intracellular amino acid levels in an in vitro myosteatosis model, differentiated PHMs grown on 96-well plates were washed thrice with amino acid-free DMEM and incubated in DMEM containing amino acid concentrations that matched those found in healthy human plasma[61] (see Table S10). For lipotoxic stimulation, this same medium was used with the addition of 200 μM free fatty acids (2:1 oleate:palmitate), 19 μM L-glucose, and 10 ng/mL TNF-α (Thermo Fisher). After 24 h of equilibration in these media, cellular metabolites were extracted to measure baseline amino acid levels. In parallel, cells were incubated with the constituent amino acids of LIVQNacHKFT at specified fold concentrations above plasma levels (see Table S10). N-acetylcysteine (Nac) is not endogenous in plasma and was proportionally scaled to be 0.2 or 0.5 mM. Following this treatment, cellular metabolites were extracted at 0.25-, 0.5-, 1-, and 2-h time points for amino acid analysis.

**RT-PCR**. Total RNA was isolated from cell lines using the RNeasy Mini Kit (Qiagen). First strand cDNA was synthesized using 2 μg RNA and SuperScript II reverse transcriptase (Invitrogen), according to the manufacturer's

recommendations. PCR was performed over 30 cycles, using the *Taq* DNA polymerase kit (Qiagen) with primers listed in Table S11.

**RNA silencing**. Silencing of B⁰AT2 in U87-MG cells was performed using Ambion Silencer Select predesigned siRNA ID: s224342. On the day before transfection, cells were split and seeded out in 35 mm cell culture dishes at 150,000–300,000 cells. Immediately before transfection the medium was renewed. For transfection (all volumes per dish), 4 μL of Lipofectamine RNAimax (Life Technologies) was combined with 250 μL OPTI-MEM (Life Technologies) and separately 30 pmol of siRNA was combined with 250 μL of OPTI-MEM. Both solutions were combined after 5 min and were incubated for a further 20–30 min at room temperature before adding the transfection complexes dropwise to the cells. Transfected cells were incubated at 37 °C and 5% CO₂ for 4–6 h after which the medium was replaced with fresh DMEM/Ham's F12/FBS10%/2 mM glutamine. Universal negative control (Sigma Aldrich, SIC001) was used as a scramble control.

**Surface biotinylation and western blotting**. For surface biotinylation, cells were washed three times with modified PBS (0.6 mM MgCl₂ and 1 mM CaCl₂, pH 8.0) and subsequently incubated with 1 mg/mL Sulfo-NHS-LC-Biotin (Thermo Fisher) in modified PBS for 30 min. Excess biotinylating agent was quenched and removed with three washes of PBS supplemented with 0.1 M glycine and cells were homogenized using a lysis buffer (150 mM NaCl, 20 mM Tris, and 1% Triton X-100, pH 7.5) containing protease inhibitors (Roche). Lysates were incubated for 2 h on ice with occasional mixing and then cleared by centrifugation. Protein concentration was determined by the Bradford assay to normalize the mass of protein added to high-capacity streptavidin agarose resin (Thermo Fisher). After overnight incubation with slow rotation at 4 °C, unbound protein was removed by five washing cycles of the resin with lysis buffer. Biotinylated proteins were mixed with 4× LDS sample buffer and were disassociated from the resin using sample reducing reagent (Thermo Fisher) and by incubation at 90 °C for 5 min. Proteins were then separated on 4–12% Bolt Bis-Tris gels (Thermo Fisher) for 50 min and transferred to a nitrocellulose membrane (GE Healthcare). Membranes were blocked in 5% (w/v) skim milk powder in PBS with 0.15% Tween 20 overnight, before being incubated with primary and secondary antibodies, as listed in Table M5. Densitometry of protein bands was performed using Image J[62]. Uncropped images of all blots are provided in the Source data file associated with this manuscript.

**Amino acid flux assays**. Amino acid transport velocities of cells were determined by measuring the uptake of radiolabeled amino acids during the linear range of accumulation (2 min for leucine, alanine, and glutamine, and 6 min for all other substrates). Experiments were conducted on cells grown to confluence in 35 mm dishes at 37 °C. A combination of Hanks' balanced salt solution (HBSS; 137 mM NaCl; 5.4 mM KCl; 2.7 mM Na₂HPO₄; 1 mM CaCl₂, 0.5 mM MgCl₂; 0.44 mM KH₂PO₄; 0.4 mM MgSO₄; 5 mM D-glucose, 5 mM HEPES; pH 7.4) and HBSS where sodium was substituted with lithium or NMDG, were used to discriminate between transporters. Amino acid transporter inhibitors and competing substrates were also used to further parse the activities of transporters with similar functional characteristics.

Briefly, cells were washed with the appropriate HBSS solution prior to adding HBSS containing a given amino acid in radiolabeled (all purchased from PerkinElmer, ≥2000 cpm/nmol) and unlabeled (100 or 300 μM) forms, and in the presence or absence of inhibitors and competing substrates. After two or 6 min of incubation at 37 °C, cells were washed with ice-cold HBSS and lysed with 0.1 M HCl. Cells were scraped and homogenized, with one portion of the homogenate transferred to a scintillation vial for counting, while the other was used for protein determination to normalize the rates of uptake. Concentrations of inhibitors and incubation time are listed in Table S1 (Supplementary material).

**Oocyte expression systems and flux experiments**. *Xenopus laevis* oocytes were isolated and maintained, as described previously[63]. Surgery and oocyte removal complied with the Australian Code for the Care and Use of Animals for Scientific Purposes and the ACT Animal Welfare Act 1992, and were approved by the Animal experimentation ethics committee of the Australian National University under the reference numbers A2020/28 and A2017/36. Selected oocytes were injected with 10 ng of rat EAAT1 and human ASCT2, and were used as described previously[64]. Experiments were performed after 4 days of expression. To measure efflux, oocytes were preloaded with 100 μM [¹⁴C]glutamate for 1 h. After washing three times with ND96 (96 mM NaCl, 2 mM KCl, 1 mM MgCl₂, 1.8 mM CaCl₂, and 5 mM HEPES-NaOH, pH 7.4), efflux was initiated by addition of 0.5 mM glutamine to the ND96 incubation buffer. Controls remained without addition. Recapture of glutamate was inhibited by the glutamate transport inhibitor UCPH-101 (10 μM).

**Cell volumetry**. The volumes of A549 and U87-MG cells were measured using a Multisizer 4 (Beckman Coulter) with a 100 μm aperture tube. Prior to volume measurement, cells were trypsinized, pelleted, and resuspended in calcium-free HBSS. This same solution was used as the electrolyte solution for the aperture tube. Approximately 5 × 10⁵ cells were diluted into 10 mL of calcium-free HBSS in a cuvette. Cell volume measurements were taken over 60 s, and distributed in 200

bins spanning a volume range of 500–6000 fL. A lognormal fit was applied to the volume distribution data to determine an approximation of the mean cell volume for each cell line. To determine the volume of myotubes the same treatment paradigm as described under cell culture was used. On day 8, after 24 h of equilibration in custom media, cells were washed once in PBS, fixed in 4% paraformaldehyde, and permeabilized with 0.5% Triton X-100. Cells were then stained with Cell Mask (Thermo, H32721) at 2ug/mL for 30 min, followed by three washes of PBS. Molecular Devices Confocal Image Express High Content Screening platform was used to acquire z-stacking images spanning all cell layers. Custom image analysis module was applied to z-stacking images to determine cell volume based on cell mask stain. Cell volumes for ten wells were measured and final cell volume is calculated as average of those ten-well replicates. For total protein quantification in myotubes, cells were harvested after 24 h of equilibration in custom media, lysed with RIPA buffer (Cell Signaling Technology), then followed by BCA assay (Sigma) according to manufacturing instructions.

**Amino acid equilibration experiments and LC-MS.** To study changes in cellular amino acid equilibria, A549 and U87-MG cells were incubated in BME with 1 mM pyruvate and nonessential amino acids, including glutamine at 0.6 mM, or a modified HBSS solution (supplemented with 23 mM NaHCO3 and with all proteinogenic amino acids at a baseline concentration of 100 μM). Cells were initially seeded at ~200,000 cells per 35 mm dish. Once confluent (2 days post-seeding for A549 cells and 5 days for U87-MG cells), cells were washed and incubated in the media for 30 min at 37 °C. Each dish was then washed once with ice-cold 0.9% NaCl solution followed by a second wash with ice-cold Milli-Q water, before immediately snap freezing the cells in liquid nitrogen, adding 600 μL of extraction solution (60:40 methanol:water) with isotopically labeled amino acid internal standards (MSK-A2-1.2 and CLM-1822-H-0.1, Cambridge Isotope Laboratories, all at 1.5 μM), and placed on dry ice. Cells were then thawed, collected, and mixed with an equal volume of chloroform. The lysate was vortexed for 5 min followed by another 5 min of sonication and the aqueous phase was cleared by centrifugation, transferred to a new microcentrifuge tube, and placed in a vacuum concentrator for 3 h. Metabolites were resuspended in acetonitrile:10 mM ammonium acetate (90:10 + 0.15% formic acid) for LC-MS analysis. This method was also used to measure glutaminase activity in A549 and U87-MG cells using CB-839 (10 μM), with the exception of the incubation time being extended to 60 min. Repeated washing steps did not change the recovery of amino acids (Table S13). A final wash in with ice-cold Milli-Q water is an established step for LC-MS analysis to reduce ion suppression[65]. Glutamine-derived glutamate efflux was measured by washing cells grown to confluence in 35 mm dishes twice with BME, and incubating them in supplemented BME with 1% dialyzed FCS and 1 mM [13C-5] L-glutamine in the presence or absence of L-GPNA (3 mM) and erastin (10 μM) at 37 °C. Over a period of 24 h, 15 μL of the medium was sampled at various time points and were prepared for LC-MS analysis, as detailed above.

Amino acids were analyzed with an Orbitrap Q-Exactive Plus coupled to an UltiMate 3000 RSLCnano system (Thermo Fisher). Separation of analytes was achieved using a 3 μm ZIC cHILIC 2.1 × 150 mm column (EMD Millipore) with a gradient elution of two mobile phases over 21 min at a flow rate of 0.4 mL/min. Solvent A was comprised of 10 mM ammonium acetate in water with 0.15% formic acid and solvent B consisted of acetonitrile with 0.15% formic acid. Solvent A was held at 10% from 0 to 6.0 min, linearly increased to 15% between 6.0 and 6.1 min, increased again to 26% from 6.1 to 10.0 min and then to 36% from 10.0 to 12.0 min and to 64% from 12.0 to 12.1 min, where it was held until 16.0 min. From 16.0 to 16.1 min, solvent A was decreased to 10% and held until 21 min. Analytes were ionized in positive mode, and analyzed with a combination of survey scans (73–194 $m/z$, $R = 35,000$) and parallel reaction monitoring ($R = 17,500$), both with the automatic gain control target set to $5 \times 10^4$ charges, a maximum injection time of 50 ms and an isolation window of 0.5 $m/z$. Collision energies were optimized for each analyte (see Table S14). Quantification was achieved using Xcalibur (Thermo Fisher) by integrating the base peak in the MSMS spectra of all amino acids and their isotopes, except glycine, alanine, and aspartate (due to the production of daughter ions that were below the $m/z$ instrument detection threshold or unreliable fragmentation patterns) for which parent ion peaks were selected for quantification (see Table M5). Tryptophan and asparagine could not be quantified due to the absence of their respective isotopes in the internal standards mixture, and cysteine was omitted from analysis due to its high intrinsic reactivity. Cytosolic concentrations were determined according to the following equation:

$$[AA]_{cyt} = \frac{[AA]_{vial} \times V_{vial}}{V_{cell} \times m_{protein} \times c}$$

Where $V_{cell}$ is the mean cell volume, $m_{protein}$ is the mean protein mass per dish measured in satellite dishes in each experiment, and $c$ is the correlate of protein mass and cell number (7036 ± 488 cells/μg for A549 cells and 7092 ± 830 cells/μg for U87-MG cells). This correlate was determined by trypsinizing, pelleting, and resuspending A549 and U87-MG cells in HBSS, measuring cell density with a hemocytometer and measuring protein mass using the Bradford assay.

To extract amino acids from cultured myotubes, cells were washed once with cold PBS, and then quenched with 100 μL ice-cold extraction solvent (80:20, methanol:water) containing isotopically labeled internal standards (MSK-CAA-1, Cambridge Isotope Laboratories, all at 1 μM). Five wells of lysates from same treatment condition were pooled into Eppendorf tubes. Lysates were incubated at −20 °C for 30 min, followed by at least 2 h at −80 °C, and cleared by centrifugation. Supernatants were transferred to new sets of Eppendorf tubes, desolvated in a speed vacuum concentrator, and resuspended in 20 μL of buffer composed of 42.5% acetonitrile, 57.4% water, 0.1% formic acid, and 10 mM ammonium formate. Samples were injected into a 1.7 μm, 2.1 × 150 mm Acquity UPLC BEH amide column (Waters) connected to an Ultimate 3000 UHPLC (Dionex) coupled to a Q-Exactive mass spectrometer equipped with a HESI probe (Thermo Fisher). The liquid chromatography method consisted of flow rate of 0.4 mL/min and a gradient elution of solvent A (95:5 acetonitrile:water + 0.15% formic acid) and solvent B (water + 0.15% formic acid) over a run time of 10 min per sample. Solvent A was held at 100% from 0 to 1.0 min, followed by a linear gradient to 90% solvent A between 1.0 and 3.0 min, 80% solvent A from 3.0 to 4.3 min, and 60% solvent A from 4.3 to 5.6 min. Solvent B was set to 100% from 5.6 to 7.5 min and was returned to 0% between 7.5 and 10.0 min. The full scan range of the Q-Exactive mass spectrometer was set at 50 to 300 $m/z$ with a resolution of 70,000 at 200 $m/z$. The maximum injection time was 100 ms and the automated gain control was targeted at $3 \times 10^6$ ions. Amino acids were quantified using TraceFinder software (Thermo Scientific) by measuring the area under the peak for the targeted endogenous analyte in comparison to the isotopically labeled internal standard. Note that this analytical approach can detect most proteinogenic amino acids, but cannot reliably measure Nac or cysteine, due to high intrinsic reactivity and instability for these analytes.

**Stable isotopic tracing of glutamine and GC-MS analysis.** U87-MG cells were seeded onto 35-mm dishes and incubated in growth medium until they reached confluence, washed thrice with warm HBSS, and incubated with modified HBSS (23 mM NaHCO3) containing 1 mM of [13C-5]L-glutamine (Cambridge Isotope Laboratories). Metabolites were extracted at 0.3-, 1-, 2-, and 24-h time points by aspirating HBSS, washing cells once with 0.9% NaCl, and depositing enough liquid nitrogen to cover the surface of each dish. Cells were quenched by the addition of 600 μL of ice-cold extraction solvent (9:1 methanol:chloroform) and 10 μL of internal standard (ribitol; 10 mM) to each dish, and were incubated on dry ice for 10 min. Cells were then scraped and collected into microcentrifuge tubes and incubated on ice for 5 min. Lysates were cleared by centrifugation for 5 min at 4 °C, before transferring 150 μL of the supernatant into autosampler vials and dried in a speed vacuum concentrator for 4 h. Quality control samples were also prepared by pooling equal volumes of each sample to monitor sample stability and analytical reproducibility throughout each batch sequence run. Derivatization was carried out by a robotic Gerstel MPS2 multipurpose sampler in which dried extracts were incubated at 37 °C for 90 min at 750 r.p.m. after adding 20 μL of pyridine containing 20 mg/mL methoxyamine hydrochloride (Supelco). After this, 25 μL of N-(tert-butyldimethylsilyl)-N-methyltrifluoroacetamide and 1% (w/v) tert-butyldimethylchlorosilane (MTBSTFA + 1% TBDMCSI; Sigma Aldrich) was added to the samples, which were incubated for a further 60 min. Five microliters of a mixture of n-alkanes was also added to the samples before injection into the GC-MS instrument. One microliter of each sample was injected in split and splitless mode in an Agilent 7890 A gas chromatograph coupled to an Agilent 5975 C single quadrupole mass spectrometer. The Agilent 7890 A was equipped with a Varian factor four VF-5 capillary column (30 m long, 0.25 mm inner diameter, and 0.25 μm film thickness). The injector temperature was set to 230 °C and helium was used as a carrier gas at a flow rate of 1 mL/min. Oven temperature was initially held at 100 °C for 1 min and increased to 270 °C at rate of 7 °C per minute, then rising to 300 °C at a rate of 10 °C per minute and holding at this temperature for 1 min. The total run time was therefore 29 min per sample. A solvent delay was also added to the method until 7.1 min. The electron impact ion source was kept at 260 °C and filament bias at −70 eV. The mass spectrometer was operated in full scan mode ranging from 40 to 600 $m/z$ using a scan rate of 3.6 Hz. The fractional labeling of the carbon atoms was determined from the mass isotopomer distributions of the following fragments: malate ($m/z$ 233, C1–C4), aspartate ($m/z$ 302, C1–C4), glutamate ($m/z$ 432, C1–C5), and glutamine ($m/z$ 431, C1–C5). The mass isotopomer distributions were corrected for natural abundance using the program Data Extraction for Stable Isotope-labeled metabolites (DEXSI) software package[66].

**Measuring the rate of leucine incorporation in protein.** To measure the effects of protein synthesis on cytosolic amino acid equilibria, A549 and U87-MG cells were grown to confluence in 35 mm dishes, washed twice with BME, and incubated with supplemented BME (see cell culture) with [14C-U] L-leucine (diluted 1:2650). At various time points, cells were washed twice with ice-cold HBSS and lysed with 0.1 M HCl. Lysates were collected in a microcentrifuge tube, to which 100 μL of trichloroacetic acid was added before vortexing and placing on ice for 30 min. Samples were then pelleted in a microcentrifuge (maximum speed for 10 min at 4 °C). Supernatant was removed and the protein pellets were washed once with ice-cold acetone before drying in open air for 10 min. The isolated protein samples were then hydrolyzed in 50 μL 2 M KOH at 70 °C for 30 min. Samples were neutralized with the addition of 100 μL 1 M HCl, diluted in 350 μL of Milli-Q water, and prepared for scintillation counting. The Bradford assay was conducted on satellite dishes to normalize rates to total cell protein.

**Derivation of parameters for computer simulation.** Substrate affinity ($K_M$)-values were taken from Table S3. The table shows a curated list of $K_M$-values, which in the opinion of the authors represent the most reliable experimental values. Many studies did not determine the $K_M$ of all possible substrates, in which case an estimate (est) was made based on physical relation to known substrate $K_M$-values, general trends of $K_M$-values observed for a particular transporter and the ability of amino acids to compete with a key substrate. Transport rate of individual transporters ($v$) were taken from Table S6: experiments to quantify the rate of individual transporters are outlined in the text accompanying Figs. 2 and 3. Conditions for all possible transporters are outlined in Table S1. To calculate $V_{max}$ the following equation was used:

$$V_{max} = \frac{v(K_M + [S])}{[S]}$$

In the table, the transport rate $v$ was normalized to 1 min before calculation of $V_{max}$.

**Statistics and reproducibility.** In all experiments, "$n$" refers to the total number of biologically independent samples analyzed in the experiment. The number of independent experiments used to generate the samples is given as "$e$". One-way ANOVA was used to compare samples taken under different conditions. To compare two samples in a larger group Tukey's multiple comparison test was used. The match between simulation and in vitro experiments was analyzed by linear regression after plotting results from in vitro experiments on the ordinate and simulation results on the abscissa. Pearson's correlation coefficient was used to report the goodness of fit. To analyze changes, quadrant plots were used. In these plots, the difference between the starting concentrations ($t = 0$) and the chosen time point was calculated. The absolute values were used to calculate $\log_2$ after which the signs were reinstated. The match between simulation and in vitro experiments was analyzed by linear regression after plotting $\log_2$ differences from in vitro experiments on the ordinate and the corresponding simulation results on the abscissa. Pearson's correlation coefficient was used to report the goodness of fit.

**Images.** Schematic images were created with Biorender.com.

**Reporting summary.** Further information on research design is available in the Nature Research Reporting Summary linked to this article.

## Data availability
Data are also accessible in Mendeley Data: B.S.; G.-C.G.; V.J.; Z.Z.; C.W.; X.S.; J.K.; B.A. (2021), "Quantitative modeling of amino acid transport and homeostasis in mammalian cells", Mendeley Data, V1, https://doi.org/10.17632/6k3k4hcftt.1. Source data are provided with this paper.

## Code availability
The code for JDFC is available through Github as JDFC_v1.1. Accession: stefanbroeer/JDFC_v1.1: Updated version of JDFC (github.com).

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

## Acknowledgements

The development of JDFC went through many earlier versions using different platforms. The authors would like to thank Jue Sheng Ong (QIMR Berghofer Medical Research Institute), Ben Corry (ANU Research School of Biology), and Damien Hall (Research School of Chemistry) for their initial attempts to develop transport simulation, and Avinash Upadhya and Woei Ming (Steve) Lee (ANU John Curtin School of Medical Research ) for considerable help with Matlab scripting. Adam Carroll and Thy Truong (ANU College of Science Joint Mass Spectroscopy facility) helped with developing the amino acid analysis. The authors thank Axcella Health team members Murat Cokol for his help with data modeling and statistical analysis. We thank Michael Hamill and Meredith Duffy (Axcella Health) for their enthusiastic support of this study. We are indebted to Lon J. Van Winkle (Rocky Vista University) and Reinhard Krämer (University of Cologne) for commenting on drafts of the manuscript. Work associated with this project was funded by Australian Research Council Grant DP180101702.

## Author contributions

Conceptualization: S.B.; methodology: G.G.-C., K.J. and S.X.; software: J.V. and Z.Z.; validation: G.G.-C., K.J., A.B. and W.C.C.; formal analysis: W.C.C., S.B., G.G.-C., J.V. and S.X.; investigation: G.G.-C., A.B., W.C.C. and K.J.; writing: G.G.-C. and S.B.; editing: all authors; supervision: S.B.; project administration: S.B. and B.C.; and funding acquisition: S.B.

## Competing interests

AXA2678 is an amino acid mixture developed by Axcella Health. W.C.C. and S.X. are owners of Axcella Health stock and stock options. The remaining authors have no competing interests.
