## [Peer Review File · Nature Communications]

Quantitative modelling of amino acid transport and homeostasis in mammalian cellsREVIEWER COMMENTS

Reviewer #1 (Remarks to the Author):

The manuscript by Broer's lab is an excellent piece of work that for the first time approaches amino acid transport in mammalian cells at a systems-level. The approach is simple and brilliant: i) database and their own qualitative mRNA expression data, together with surface expression of transporters revealed the transporters present in the two selected cultured cells. ii) Quantification of transport flux for each transporter using up-to-date strategies (selection of model substrates, specific inhibitors), and iii) using available K_m values at both sides of the plasma membrane and iv) quantifying the cell content of amino acids allow the authors to: i) quantify the contribution of each transporter for the transmembrane flux for each amino acid, and ii) model the cytosolic concentration of each amino acid. To validate their modelling, the authors used spiked increments of several single amino acids and blocking LAT1 and SNAT1 with specific inhibitors. Finally, this work also modelled the evolution of the cytosolic content of amino acids in myotubes upon supplementation with a mixture of amino acids (AXA2678). This time, only mRNA levels of the amino acids were used to model this system. This work, in addition to construct a model of amino acid fluxes across of the cell membrane and amino acid content in the cytosol, conclude that amino acid transporters are the main regulators of the amino acid concentration in the cytosol. The authors only identify one exception, the relevant impact of glutaminolysis and conversion of glutamate to aspartate via TCA that control glutamate and aspartate concentrations. I strongly support publication of this work in Nat Commun after minor changes and clarifications.

Minor points and clarifications.

- 1) Fig. 5A shows the flow of the modelling. Reading the main part of the text is not clear when and how metabolic conversion of amino acids has been calculated and used. Only in Fig. 5C is shown the impact of the inhibition of glutaminase in the measured concentrations of amino acids, with a clear increment in the content of Glu and Asp. Has been conversion of amino acids into protein synthesis, contribution of protein degradation and metabolization of amino acids in the intermediate metabolism has been considered? It is not clear this point for other amino acids in addition to Glu and Asp. If metabolism of other amino acids has not been considered, why the modelling is so accurate? I suggest to clarify these points.
- 2) The qualification of amino acid transporters as loaders, harmonizers, controllers and recues transporters is brilliant because explain the function of all these types of transporters to fulfil the physiological needs of the cells. I have a doubt. Why γ -LAT2 is considered a harmonizer and not a controller? This transporter is a harmonizer as an antiporter but the amino acid exchange is coupled to the chemical gradient of sodium (AA^+ in exchange with AA^0 plus Na^+). This mechanism is analogous to that of SNAT3, defined here as a controller. I suggests clarifying this point.
- 3) Still in relation with γ -LAT2; why this transporter is considered an importer of AA^+ ? I am not doubting of this conclusion because there is evidence that the very similar γ -LAT1 acts as an importer of AA^+ in non-polarized cells (e.g., macrophages). I suggest clarifying that the modelling is the source of data that support this concept. Is it possible to block specifically γ -L transporters?
- 4) At the bottom of page 6 it is stated that glutamate levels were largely unchanged, except when Gln was spiked. Could these author check this point because I do not see this in Fig. 6A.
- 5) At least when I printed the manuscript, Figs 6G and 6H are very confusing with the labels of the indicated amino acids in the plot superposing over the dots of data.

Manuel Palacín

Reviewer #2 (Remarks to the Author):

The present manuscript addresses the question of amino acid homeostasis mechanisms in mammalian

cells, providing evidence for transporters to play a major role in determining this. Specifically, the authors dissect transport mechanisms through chemical inhibitors and cellular transport assays in a few cell lines, and use the associated data to support quantitative modeling efforts that show meaningful predictive power. The authors also present the conceptual framework of considering transporters as loaders, harmonizers, and controllers, which I found useful. I have the following minor suggestions for improvement:

1. The title is not fully supported as the manuscript does not consider intracellular transamination reactions, regulators of protein synthesis/catabolism, or compartmentation. Instead of claiming a “unified model of amino acid homeostasis” perhaps the authors would be comfortable with something like “Quantitative modeling of amino acid transport and homeostasis in mammalian cells”?
 2. In terms of the model, it would be good to have a readily accessible table of all the parameter values. Perhaps I missed this in the supplement, but it seems essential enough to put the table in the methods or maintext. Also, I was confused about how the parameters of the model were determined – were they purely from literature with no manual or computational fitting? Or were values optimized within literature ranges by manual tinkering? Or was there an overt computational optimization step?
 3. Also, it seems that the differential equations were solved at steady state by running forward time steps until steady-state. Presumably, the same answers can be obtained by solving the algebraic form of the equations (with $dM/dt = 0$)? Can the authors please do this?
 4. The abstract focuses on the pipeline not any scientific insights. Perhaps a few more detailed sentences about specific learning, including the concepts of loaders, controllers, and harmonizers, time scales, fast and slow transported amino acids, etc could be included?
 5. In terms of the measurement, it seems like different extraction and measurement protocols were used for different cell types? Can the authors make some assessment of amino acid leakage during the different washing processes?
 6. Experiments are limited to pharmacology with no use of modern genetics. This should be clearly acknowledged as a limitation, preferably including in the abstract, as a similar study using genetics is still much needed.
 7. The figures are not very polished by the standards of Nature family journal. For example, I found Fig. 1 descriptive with no clear message. Fig 2 and 3 have interesting results, but they could be graphically highlighted better. Fig 4 – 7 are visually stronger, although I found 4A/B hard to read (and again descriptive) (stacked bars might work better than pie graphs) and 6E/F it is hard to align the lines with the legend.
 8. The claim that the individual amino acid perturbations extracellularly mainly impact their own intracellular concentration is interesting, but does not align well with the concept of harmonizers. Can the authors please explain why (within the context of the model) harmonizers don’t spread these changes more broadly over different intracellular amino acids? Also, can the authors be more explicit about whether any other intracellular amino acids change systematically/significantly, even if only to a lesser extent?
 9. One of my favorite results, which could perhaps be given more emphasis, is the very clear impact of sodium on the transport of most but not all amino acids. Can the authors clarify exactly how these experiments were performed, in terms of when and how sodium was removed from the medium?
- Overall, this paper is notable for blending valuable amino acid transport experiments with advances in quantitative modeling and conceptual thinking about amino acid transport and homeostasis. I definitely enjoyed the paper.

Josh Rabinowitz

Reviewer #3 (Remarks to the Author):

The authors conduct a systems level analysis of amino acid transport in two cell lines and primary myotubes. They 1) establish absence/presence of transporters, 2) estimate v_{max} experimentally, and 3) develop a metabolic model limited to amino acid transport. To this reviewer’s knowledge this study is the first systems level kinetic model of amino acid uptake, and the authors seem to show that

it is sufficient to consider amino acid transport and disregard metabolism to explain cytosolic concentrations. If absence/presence of transporters is indeed sufficient to explain amino acid concentrations, this would be a useful result not only for the field of metabolism but also signaling. The authors observe some fluxes that conflict with the established literature (e.g. glutamate excretion), however the authors do attempt to reconcile this with some experiments and discussion. It may, however, potentially also be warranted to weaken the conclusion somewhat e.g. "most amino acids can be described by transport". The experimental work seems thorough, however, there is insufficient explanation of the underlying assumptions and data processing as well as improvement potential for the data presentation and simulations that may be addressed by a major revision. See specific comments below.

Major comments:

The maximum activity (v_{max}) of a transporter depends on both the turnover number (k_{cat}) and the concentration in the membrane ($[E]$). Both the mRNA analysis and the westerns are attempts to quantify differences in $[E]$. This may be sufficient for determining absence/presence of a transporter, however, since in principle k_{cat} could differ by orders of magnitude, it may not be appropriate to compare absolute level of mRNA or protein between different transporters, a value that is considered high for one may be considered low for another. The Human protein atlas (proteinalas.org) has tabulated mRNA values for a large number of cell lines including A549 and U87-MG, perhaps it would be more informative to compare the level of each transporter to the level in all other cell lines e.g. by its z-score. It is also not made clear exactly how the information from this survey is used in the remaining study, presumably transporters that are considered absent are not considered in the v_{max} study and blocked in the metabolic model, but how are transporters with conflicting evidence treated? Have alternative explanations been considered for instances where there is no detectable mRNA but detectable protein, e.g. issues with antibody specificity? Further, the criteria for the + system in figure 1 is not clearly explained, does it correspond to the 3 different types of measurements? In such case what cutoffs are used? The figure legend also states that high mRNA expression correlates with western blotting, however, no such correlation is calculated or shown. Some transporters are marked as "inducible" with no further explanation, how is this determined?

The assumptions underlying the experimental v_{max} determination from figure 2,3,4 are not clear. The concentration used (100uM) is lower than K_m for many of the transporters, so it cannot safely be assumed that all transporters operate at v_{max} in this experiment, it is not clear if this has been considered when deconvoluting the contributions from the different transporters. On the other hand, in table S5 there is a distinction between activity and v_{max} , but it is not stated how v_{max} is calculated from activity. Perhaps v_{max} is here adjusted by the relation between K_m and the substrate concentration in the experiment? However, for some transporters the calculated v_{max} appears to be lower than the activity, could this be due to different units for activity and v_{max} ? There is also color coding in table S5 that is not explained in the legend, additionally it mentions that some values were excluded from the mean, what is this mean value used for and how were the values selected for exclusion?

It seems like an implicit assumption that inhibition is 100% efficient. If this is the case, how are the concentrations of inhibitors determined? What is the reason that some inhibitors are listed in supplementary table S1. E.g. NMDG but not shown in figure 2 and 3, is this because they are assumed to not be present in this particular cell line based on the results in figure 1?

It is mentioned that nonspecific transport and binding is determined by addition of unlabeled substrate and subtracted from the data. Is this amount negligible compared to the specific transport? Is there any particular benefit of subtracting this value instead of treating it as another factor in the ANOVA?

Mathematically, how are the contributions from the different transporters (shown in figure 4) determined? Is this done by summing up the factors in the ANOVA? In such case, perhaps reporting the confidence intervals may be informative for the reader.

Many cell lines excrete glutamate, alanine and glycine (See [Jain 2012] for a high throughput study of exchange fluxes that includes A549 or [Nilsson 2020] for a quantitative analysis of amino acid excretion). What could be the reason that only uptake is observed in this study? Is this a property specific to the studied cell lines? Are the cells nutrient/glutamine starved prior to the experiment? Or could there potentially be mixing between the medium and cytoplasm which does not constitute a net influx?

K_m values are collected from literature and it is stated that missing K_m values are interpolated, this cannot be correct, as interpolation assumes that there is a curve to interpolate. Perhaps the mean or median is used? There seems to be no further mentioning of interpolation in methods or supplementary material. Many of the K_m values say '(est)' which I assume means estimated, but it is not clear how these estimations were performed.

Using these K_m values and v_{max} the authors construct a metabolic model of amino acid transport based on Michaelis-Menten (MM) based kinetics. Through some optional parameters the model is also equipped to model steady state metabolic fluxes, but since this option does not seem to be used in this study, the presentation of this should perhaps be deferred to the discussion section. The authors opt to solve the model through an iterative approach. It is not made clear why this is preferred over using an ordinary differential equation solver, e.g. MATLAB's ode45.

Transporters with multiple substrates are handled by re-estimating apparent K_m s at each timestep. It seems like the equation for this (eq2) could have some unintuitive consequences, e.g. if one only considers a single amino acid at a concentration of K_m then it seems like the equation would become $K_{app} = K_m (1 + 1/1) = 2K_m$. Perhaps this is just an issue with the mathematical presentation, but the amino acids should not inhibit themselves.

To ensure compliance with the 2nd law of thermodynamics the authors set K_m and V_{max} to the same value for uniporters (principle 1). However strictly, it should be sufficient to require that the equilibrium constant ($K_{eq} = 1$ for this to hold. From the Haldane relationship ($K_{eq} = (V_f * K_{mP}) / (V_b * K_{mS})$) it follows that $V_b/V_f = K_{mP}/K_{mS}$. So only together with principle 4, that v_{max} is equal for the forward and reverse reaction does it follow that K_{mP} must be equal to K_{mS} . It is also explicitly required (principle 3) that the flux is zero when the metabolite is absent, it is not clear why this does not follow directly from the equations that would involve a multiplication with zero concentration.

The model predicts amino acid concentrations in good agreement with measured values. However, since the data spans orders of magnitudes, the model could in principle be correct with regards to the typical concentration of each amino acid, but at the same time be incorrect with regards to changes between cell lines, e.g. predicting a 10% increase instead of a 10% decrease would not affect the overall correlation between experimental data and model prediction. A correlation of the change would address this, this type of analysis is performed for myotubes with some success (figure 6E) but should also be performed for A549 vs U87-MG.

It seems like several of the discussion points, including the proposed functional classification of AA transporters, could be addressed in silico with a sensitivity analysis/metabolic control analysis (MCA) of the model. Such an analysis would provide numerical answers about the contributions of different transporters and parameters (including cell volume) to the model predictions. To facilitate this type of systematic analysis it may be advisable to restructure the source code so that the functional part becomes independent of the UI.

Since the model relies on experimentally determined v_{max} values and the metabolite levels predicted for myotubes from transcriptomics data appear to be off by a factor 100 (figure 6E and figure S6F) possibly due to the missing volume estimates, it may be a too strong conclusion that concentrations can be calculated using mRNA levels, however, it appears that the relative levels can be estimated.

Minor comments:

On line 180 it is stated that amino acid transport is "governed by" MM, but MM is just a simplified model, perhaps it would be more correct to write "can be described by".

In figure 1 it is not clear why the transporter names are repeated for each cell line.

When error bars are shown does this represent SEM or STD? It is becoming increasingly common to show the data points along with the barplot when the N is sufficiently small.

The legends of figure 2 and 3 state that the figures show a deconvolution, however the figure only shows the data, it is figure 4 that shows the deconvolution.

The number of letters above the bars in figure 2 and 3 do not always match up with the bars e.g. in 2F.

A 3d pictchart, as in figure 4 is not a usual way to convey scientific data.

In figure 4 all transporters are named by the convention "[Amino acid] Transport" apart from alanine that is named "Alanine transport".

Technically it is not sufficient to show a single instance of an equilibrium point (figure S6A) to show that a system is "inherently stable". However, from the formulation it does seem very probable that it is.

In the introduction it is assumed that uniporters equilibrate cytosolic and plasma amino acid concentrations, however, in principle a uniporter could provide net transport if it consumes ATP in the process, e.g. ABC transporters.

Several abbreviations are not defined at first use, e.g. for JDFC.

Table S4 is not a table.

It is stated in the manual that only matlab 2019, not 2020 is not supported. It is possible to get most of the code to run under matlab 2020 by adding the line `[variable] = arrayfun(@(x) str2double(x), [variable]);`

In figures similar to Figure S6B, it is useful to also show the line of identity.

In figure 6E, why is the log scale in base 2 instead of 10?

Direct quotes of other work, as done in the introduction, is not common in natural science, paraphrasing and citing is often sufficient.

References:

[Jain 2012] M. Jain, et al., Metabolite profiling identifies a key role for glycine in rapid cancer cell proliferation. *Science* 336, 1040–4 (2012).

[Nilsson 2020] A. Nilsson, et al., Quantitative analysis of amino acid metabolism in liver cancer links glutamate excretion to nucleotide synthesis. *Proc. Natl. Acad. Sci.* 117, 10294–10304 (2020).

REVIEWER COMMENTS

All reviewers: To assist with the substantial changes made to the manuscript a markup version of the manuscript, methods and supplementary tables has been uploaded together with the clean manuscript files.

Reviewer #1 (Remarks to the Author):

The manuscript by Broer's lab is an excellent piece of work that for the first time approaches amino acid transport in mammalian cells at a systems-level. The approach is simple and brilliant: i) database and their own qualitative mRNA expression data, together with surface expression of transporters revealed the transporters present in the two selected cultured cells. ii) Quantification of transport flux for each transporter using up-to-date strategies (selection of model substrates, specific inhibitors), and iii) using available Km values at both sides of the plasma membrane and iv) quantifying the cell content of amino acids allow the authors to: i) quantify the contribution of each transporter for the transmembrane flux for each amino acid, and ii) model the cytosolic concentration of each amino acid. To validate their modelling, the authors used spiked increments of several single amino acids and blocking LAT1 and SNAT1 with specific inhibitors. Finally, this work also modelled the evolution of the cytosolic content of amino acids in myotubes upon supplementation with a mixture of amino acids (AXA2678). This time, only mRNA levels of the amino acids were used to model this system.

This work, in addition to construct a model of amino acid fluxes across of the cell membrane and amino acid content in the cytosol, conclude that amino acid transporters are the main regulators of the amino acid concentration in the cytosol. The authors only identify one exception, the relevant impact of glutaminolysis and conversion of glutamate to aspartate via TCA that control glutamate and aspartate concentrations.

I strongly support publication of this work in Nat Commun after minor changes and clarifications.

Response: The authors thank the reviewer for the positive and constructive comments about our study.

Minor points and clarifications.

1) Fig. 5a shows the flow of the modelling. Reading the main part of the text is not clear when and how metabolic conversion of amino acids has been calculated and used. Only in Fig. 5c is shown the impact of the inhibition of glutaminase in the measured concentrations of amino acids, with a clear increment in the content of Glu and Asp. Has been conversion of amino acids into protein synthesis, contribution of protein degradation and metabolization of amino acids in the intermediate metabolism has been considered? It is not clear this point for other amino acids in addition to Glu and Asp. If metabolism of other amino acids has not been considered, why the modelling is so accurate? I suggest to clarify these points.

Response: For most amino acids transport is significantly faster than metabolism. Transport equilibrium is typically reached within 20-30 min. Metabolism and protein synthesis change amino acid concentration on a slower time-scale. If amino acids are used up by metabolism and protein synthesis, they are refilled by transport, thus levels remain the same. We can increase metabolism/conversion in the simulation, but it has very little impact on the cytosolic amino acid concentration. Apart from the conversion of glutamine to glutamate, we have not used any other metabolic reaction for the modelling. Transport is measured in basic salt solution during which no

cell growth occurs. We have measured leucine incorporation into protein in Basal Medium Eagle and even under growth conditions the incorporation is only 1% of the transport rate. These data are now presented in Fig.S6.

To address this point, we write:

Results: Incorporation of amino acids into protein was much smaller than active transport. For instance, leucine was incorporated at a rate of 0-0.4 nmol in 2 min in BME medium in A549 cells compared to a transport of 40 nmol in 2 min (Fig. S7a), the incorporation in U87-MG cells was below significance (Fig. S7b). As a result, we ignored incorporation into protein.

Discussion: Transporters determine amino acid equilibrium concentrations because uptake and efflux are much faster than metabolism and protein synthesis. During transport experiments in uptake buffer, cell growth and net amino acid incorporation into protein was negligible. If an amino acid is metabolised at a low rate inside the cell, it will be replaced by medium amino acids. A significant impact of metabolism was observed due to the conversion of glutamine to glutamate to maintain intracellular glutamate levels against active efflux via xCT and ASCT2. As observed in the amino acid spike experiments, transport processes will generate a mirror image of elevated plasma amino acid concentrations in the cytosol. After a meal, amino acid levels increase, and this will be translated into elevated cytosolic concentrations, which in turn activates metabolism. Over several hours this will remove amino acids thereby normalising cytosolic and plasma amino acid concentrations. Once fasting levels are reached, metabolism ceases. At any time, transport processes hold the ratio between cytosolic and plasma amino acid concentrations constant. When there is uneven use of amino acids for protein synthesis in the cytosol the harmonizing process will re-establish concentrations that mirror blood plasma (or media). This process also explains that glycine is released in slowly growing cancer cell lines, while it is consumed in fast growing cell lines.

2) The qualification of amino acid transporters as loaders, harmonizers, controllers and recues transporters is brilliant because explain the function of all these types of transporters to fulfil the physiological needs of the cells. I have a doubt. Why γ +LAT2 is considered a harmonizer and not a controller? This transporter is a harmonizer as an antiporter but the amino acid exchange is coupled to the chemical gradient of sodium (AA⁺ in exchange with AA⁰ plus Na⁺). This mechanism is analogous to that of SNAT3, defined here as a controller. I suggest clarifying this point.

Response: To address this point we have performed sensitivity analysis by removing single transporters from the simulation and recording the resulting changes of intracellular amino acids. The results are now shown in Fig. 7b/c. The analysis confirms the classification of γ +LAT2 as a harmonizer. Like LAT1 which performs tertiary active transport together with SNAT1/2, γ +LAT2 performs tertiary active transport in combination with CAT1. γ +LAT2 is different to SNAT3, because regarding amino acids it is an obligatory antiporter. This is confirmed by the sensitivity analysis showing an increase of amino acid concentrations upon its removal. Moreover γ +LAT2 does not really use the Na⁺ electrochemical gradient, because it can also use K⁺ cotransport (work from Rosa Deves' group cited in the manuscript).

We now write in the discussion:

Another important loader is CAT1 working in conjunction with harmonizer γ ⁺LAT2. Not all cells express CAT1 and in its absence γ ⁺LAT2 can also take up cationic amino acids as observed in some cell types, but the resulting cytosolic concentrations are low. This is consistent with a low ion selectivity of the transporter allowing it to use Na⁺ or K⁺ for cotransport.

... The classification was supported by sensitivity analysis (Fig. 7b/c). In the analysis we removed individual transporters from the simulation to observe the effect on the amino acid pool. Removal of a loader should reduce amino acid levels (shift from red to blue), while removal of a controller should increase amino acid levels (shift from blue to red). Harmonizers generate a mixed picture or change very little because the mix is already harmonized. Consistently, loss of CAT1 and SNAT1/2/4 generated a blue shift. The redundancy between SNAT1/2/4 was clearly visible in comparison to the combined removal of all three transporters. The harmonizer γ +LAT2 can lift a variety of neutral amino acids at the expense of cationic amino acids loaded by CAT1. CAT1 appears to be the dominant loader in A549 cells (Fig. 7B), while the load was more evenly spread in U87-MG cells (Fig. 7c). SNAT5 increases amino acid levels when removed, consistent with its controller function.

3) Still in relation with γ +LAT2; why this transporter is considered an importer of AA+? I am not doubting of this conclusion because there is evidence that the very similar γ +LAT1 acts as an importer of AA+ in non-polarized cells (e.g., macrophages). I suggest clarifying that the modelling is the source of data that support this concept. Is it possible to block specifically γ +L transporters?

Response: In vitro, we could silence γ +L, but this is unlikely to remove its activity completely. In the simulation, we can remove γ +L (below), which is insightful.

In the presence of γ +LAT2, CAT1 acts as an importer (grey, positive flux) and γ +LAT2 as an efflux pathway (green, negative numbers). In the absence of CAT1, net arginine efflux via γ +LAT2 quickly ceases and cytosolic arginine concentrations are maintained at a lower level. Overall, we feel that the intricacies of γ +LAT2 deserve a manuscript in its own right. There is no space in the main manuscript to incorporate these data and its details would distract from the main message of this study.

4) At the bottom of page 6 it is stated that glutamate levels were largely unchanged, except when Gln was spiked. Could these author check this point because I do not see this in Fig. 6A.

Response: This statement was indeed ill-phrased. We have rephrased to "Glutamate levels were modulated by spiking with amino acids, but there was no clear jump of cytosolic glutamate upon increasing the extracellular concentration of glutamate itself consistent with low levels of transport."

5) At least when I printed the manuscript, Figs 6G and 6H are very confusing with the labels of the indicated amino acids in the plot superposing over the dots of data.

Response: We have improved the labelling of all panels in Fig. 6 as requested.

Reviewer #2 (Remarks to the Author):

The present manuscript addresses the question of amino acid homeostasis mechanisms in mammalian cells, providing evidence for transporters to play a major role in determining this. Specifically, the authors dissect transport mechanisms through chemical inhibitors and cellular transport assays in a few cell lines, and use the associated data to support quantitative modelling efforts that show meaningful predictive power. The authors also present the conceptual framework of considering transporters as loaders, harmonizers, and controllers, which I found useful.

Response: The authors appreciate the positive summary of our study.

I have the following minor suggestions for improvement:

1. The title is not fully supported as the manuscript does not consider intracellular transamination reactions, regulators of protein synthesis/catabolism, or compartmentation. Instead of claiming a “unified model of amino acid homeostasis” perhaps the authors would be comfortable with something like “Quantitative modeling of amino acid transport and homeostasis in mammalian cells”?

The authors are comfortable with the change of title. We agree that this study focuses on amino acid transport, however, metabolism was incorporated where it was necessary to maintain intracellular amino acids at steady state levels. For most amino acids, transport is significantly faster than metabolism. Transport equilibrium is typically reached within 20-30 min. Metabolism and protein synthesis change amino acid concentration on a slower time-scale. If amino acids are used up by metabolism and protein synthesis, they are refilled by transport, thus levels remain the same. We can increase metabolism/conversion in the simulation, but it has very little impact on the cytosolic amino acid concentration. Apart from the conversion of glutamine to glutamate, we have not used any other metabolic reaction for the modelling. Transport is measured in basic salt solution during which no cell growth occurs. We have measured leucine incorporation into protein in Basal Medium Eagle and even under growth conditions the incorporation is only 1% of the transport rate. These data are now presented in Fig.S7a/b

We now write: Results: *Incorporation of amino acids into protein was much smaller than active transport. For instance, leucine was incorporated at a rate of 0-0.4 nmol in 2 min in BME medium in A549 cells compared to a transport of 40 nmol in 2 min (Fig. S7a), the incorporation in U87-MG cells was below significance (Fig. S7b). As a result, we ignored incorporation into protein.*

Discussion: *Transporters determine amino acid equilibrium concentrations because uptake and efflux are much faster than metabolism and protein synthesis. During transport experiments in uptake buffer, cell growth and net amino acid incorporation into protein was negligible. If an amino acid is metabolised at a low rate inside the cell, it will be replaced by medium amino acids. A significant impact of metabolism was observed due to the conversion of glutamine to glutamate to maintain intracellular glutamate levels against active efflux via xCT and ASCT2. As observed in the amino acid*

spike experiments, transport processes will generate a mirror image of elevated plasma amino acid concentrations in the cytosol. After a meal, amino acid levels increase, and this will be translated into elevated cytosolic concentrations, which in turn activates metabolism. Over several hours this will remove amino acids thereby normalising cytosolic and plasma amino acid concentrations. Once fasting levels are reached, metabolism ceases. At any time, transport processes hold the ratio between cytosolic and plasma amino acid concentrations constant. When there is uneven use of amino acids for protein synthesis in the cytosol the harmonizing process will re-establish concentrations that mirror blood plasma (or media). This process also explains that glycine is released in slowly growing cancer cell lines, while it is consumed in fast growing cell lines.

2. In terms of the model, it would be good to have a readily accessible table of all the parameter values. Perhaps I missed this in the supplement, but it seems essential enough to put the table in the methods or main text. Also, I was confused about how the parameters of the model were determined – were they purely from literature with no manual or computational fitting? Or were values optimized within literature ranges by manual tinkering? Or was there an overt computational optimization step?

Response: The input parameters of the model can be found in tables S3 and S6. Km-values were taken from the literature (Table S3), while Vmax-values were derived from experimental data presented in Fig. 2 and 3 and compiled in Table S6. For many transporters Km values for the main substrates are known. The remaining Km values were estimated (“est” in Table S5) from competitive inhibition plots, allowing to interpolate approximate Km-values. The dataset is too extensive to be presented in the main body of the text. The datasets were used essentially without further modification. Some adjustments were made to the Vmax of controllers, particularly SNAT5, if amino acid levels dropped too low in simulations. We feel this is justified because there is a significant error in the estimation of the Vmax of these controllers due to their low affinity. The error associated with deconvolution is now documented in revised figure 3 (previously fig. 4). A high activity of a controller will reduce the overall concentration of most neutral amino acids, while a low activity will increase their concentration. In theory this could be built as a loop to optimise parameters. In this study it was performed manually. To further address this point we have performed sensitivity analysis by removing single transporters from the simulation and record the resulting changes of intracellular amino acids. The results are now shown in Fig. 7b/c.

3. Also, it seems that the differential equations were solved at steady state by running forward time steps until steady-state. Presumably, the same answers can be obtained by solving the algebraic form of the equations (with $dM/dt = 0$)? Can the authors please do this?

Response: We have only used steady state equations in this study. Because equilibria are the product of several transporters working against or in conjunction with each other, it is not possible to solve the equations to identify the equilibrium point. However, we have tested each individual transporter equation and compared it to the thermodynamical equilibrium point as calculated by thermodynamical equations equal or similar to equations 5 and 6. We now show in table S5 expected and simulated equilibrium points for each transporter.

4. The abstract focuses on the pipeline not any scientific insights. Perhaps a few more detailed sentences about specific learning, including the concepts of loaders, controllers, and harmonizers, time scales, fast and slow transported amino acids, etc could be included?

Response: Within the tight limits of the journal instructions we have modified the abstract to read:

We show that a cell's endowment with amino acid transporters can be deconvoluted experimentally and used this data to computationally simulate amino acid translocation across the plasma membrane. Transport simulation generates cytosolic amino acid concentrations that are close to those observed in vitro. Perturbations of the system were replicated in silico and could be applied to systems where only transcriptomic data are available. The simulation explains amino acid homeostasis at the systems-level, through a combination of uniporters, symporters and antiporters, functionally acting as loaders, harmonizers and controller transporters to generate a stable equilibrium of all amino acid concentrations.

5. In terms of the measurement, it seems like different extraction and measurement protocols were used for different cell types? Can the authors make some assessment of amino acid leakage during the different washing processes?

Response: To address this point, we have performed additional experiments and measured intracellular amino acid concentration using different numbers of washing cycles. The data are presented in Table M6 of the methods section. We also provide a reference to a larger study investigating different washing procedures for MS analysis. The result demonstrates that amino acid pools are very stable during the washing procedures. The water-washing step is well-established in metabolomics to reduce the ion-suppression in LC-MS analysis.

6. Experiments are limited to pharmacology with no use of modern genetics. This should be clearly acknowledged as a limitation, preferably including in the abstract, as a similar study using genetics is still much needed.

Response: We used this strategy deliberately as pharmacological tools can be readily used by any laboratory. We have used RNA-mediated silencing in this study to verify the specificity of Loratadine inhibiting BOAT2 (Fig. S5). This figure also confirms that silencing is less complete than pharmacological inhibition. This limits the use of silencing to quantify transporter contributions. We have also used Crisp/Cas9 mediated knock-out in several studies on amino acid transport in cancer cells (doi: 10.1074/jbc.RA118.006378; and doi: 10.1074/jbc.M115.700534). These reveal significant plasticity in transporter expression to deal with deletion of harmonizers and loaders. Thus, after genetic ko the contributions are different. To the best of our abilities, the only approach to identify the correct contributions of all transporters is the pharmacological approach and we do not feel that this is a limitation.

7. The figures are not very polished by the standards of Nature family journal. For example, I found Fig. 1 descriptive with no clear message. Fig 2 and 3 have interesting results, but they could be graphically highlighted better. Fig 4 – 7 are visually stronger, although I found 4A/B hard to read (and again descriptive) (stacked bars might work better than pie graphs) and 6E/F it is hard to align the lines with the legend.

Response: We have modified the figures extensively, and moved Figure 1 to the supplemental section. We have included more analytical data arising from modelling and from responses to assessor's comments, which can be found in Figure 6 and 7. We have improved Figure 2 and 3 (now Figure 1 and 2), changed figure 4a/b (now Figure 3a/b) to stacked bar graphs and improved labelling

of figure 6 (now Figure 5). We would respectfully disagree with the notion that Fig 4a/b are descriptive, because they are the result of an extensive data analysis of amino acid uptake rates.

8. The claim that the individual amino acid perturbations extracellularly mainly impact their own intracellular concentration is interesting, but does not align well with the concept of harmonizers. Can the authors please explain why (within the context of the model) harmonizers don't spread these changes more broadly over different intracellular amino acids? Also, can the authors be more explicit about whether any other intracellular amino acids change systematically/significantly, even if only to a lesser extent?

Response: This is an important point raised by the assessor. The harmonisation process occurs across membranes and has to consider that the extracellular side (or medium) is dominant due to its large volume. This ensures that a rise of plasma amino acids is translated into a rise of intracellular amino acids, which increases metabolism to bring amino acid levels back to homeostatic levels. If there is a depletion of amino acids in the cytosol this mechanism will bring amino acid levels back to the ratios observed in blood plasma. The harmonisation becomes apparent when more than one amino acid is spiked, which we now document extensively in Fig. 5 and 6. We have also performed sensitivity analysis, by removing individual transporters in the simulation, as suggested by assessor 3. The data are shown in Fig. 7 and support the classification as harmonizers.

We write in results: *"To further analyse the response of the simulation to extracellular amino acid changes, we added a mix of five amino acids and followed amino acid changes over 1h in A549 cells (Fig. 5e/f/g). To evaluate the fit of the simulation, we used a quadrant plot of the \log_2 (+/- reassigned after log conversion) of measured (y-axis) and simulated changes (x-axis). The spiked amino acids were in the upper-right quadrant indicating an increase of the cytosolic concentrations (red dots), while non-spiked amino acids served as efflux substrates ending up in the lower left quadrant (black dots). The simulation and experimental data showed a strong correlation of $R=0.99$, $R=0.94$ and $R=0.92$ at $t=0.25h$ (Fig. 5e), 0.5 (Fig. 5f) and $1h$ (Fig. 5g), respectively. In U87-MG cells, experimental and predicted absolute amino acid concentrations showed a high level of correlation with $R=0.95$, $R=0.92$ and $R=0.89$ at $t=0.25h$ (Fig. S8c), 0.5 (Fig. S8d) and $1h$ (Fig. S8e), respectively. The changes as analysed by a quadrant plot also showed strong correlation between simulation and experiment with $R=0.98$, $R=0.95$ and $R=0.97$ at $t=0.25h$ (Fig. S8f), 0.5 (Fig. S8g) and $1h$ (Fig. S8h), respectively."*

And in the Discussion: *"The classification was supported by sensitivity analysis (Fig. 7b/c). In the analysis we removed individual transporters from the simulation to observe the effect on the amino acid pool. Removal of a loader should reduce amino acid levels (shift from red to blue), while removal of a controller should increase amino acid levels (shift from blue to red). Harmonizers generate a mixed picture or change very little because the mix is already harmonized. Consistently, loss of CAT1 and SNAT1/2/4 generated a blue shift. The redundancy between SNAT1/2/4 was clearly visible in comparison to the combined removal of all three transporters. The harmonizer γ -LAT2 can lift a variety of neutral amino acids at the expense of cationic amino acids loaded by CAT1. CAT1 appears to be the dominant loader in A549 cells (Fig. 7b), while the load was more evenly spread in U87-MG cells (Fig. 7c). SNAT5 increases amino acid levels when removed, consistent with its controller function."*

"...As observed in the amino acid spike experiments, transport processes will generate a mirror image of elevated plasma amino acid concentrations in the cytosol. After a meal, amino acid levels increase, and this will be translated into elevated cytosolic concentrations, which in turn activates metabolism. Over several hours this will remove amino acids thereby normalising cytosolic and plasma amino acid

concentrations. Once fasting levels are reached, metabolism ceases. At any time, transport processes hold the ratio between cytosolic and plasma amino acid concentrations constant. When there is uneven use of amino acids for protein synthesis in the cytosol the harmonizing process will re-establish concentrations that mirror blood plasma (or media). This process also explains that glycine is released in slowly growing cancer cell lines, while it is consumed in fast growing cell lines.”

9. One of my favorite results, which could perhaps be given more emphasis, is the very clear impact of sodium on the transport of most but not all amino acids. Can the authors clarify exactly how these experiments were performed, in terms of when and how sodium was removed from the medium? Overall, this paper is notable for blending valuable amino acid transport experiments with advances in quantitative modeling and conceptual thinking about amino acid transport and homeostasis. I definitely enjoyed the paper.

Response: The authors thank the reviewer for this positive comment. Na⁺-ions are removed immediately before addition of radiolabelled substrate to avoid depletion of amino acids, which would have an indirect effect on transport, particularly antiporters. Na⁺-dependence is an important step in the classification of transporters. While loaders are typically Na⁺-dependent, exception do exist, such as ASCT2, which despite being an antiporter is also Na⁺-dependent.

We write in results: *“Ion dependence was analysed by replacing Na⁺ with the impermeable organic cation N-Methyl-D-glucamine⁺ (NMDG) or with Li⁺ immediately before addition of radiolabelled substrate.”*

In the discussion: *“Strong Na⁺-dependence of alanine and glutamine uptake is an indication that their transport is mediated by loaders.”*

Reviewer #3 (Remarks to the Author):

The authors conduct a systems level analysis of amino acid transport in two cell lines and primary myotubes. They 1) establish absence/presence of transporters, 2) estimate v_{max} experimentally, and 3) develop a metabolic model limited to amino acid transport. To this reviewer’s knowledge this study is the first systems level kinetic model of amino acid uptake, and the authors seem to show that it is sufficient to consider amino acid transport and disregard metabolism to explain cytosolic concentrations. If absence/presence of transporters is indeed sufficient to explain amino acid concentrations, this would be a useful result not only for the field of metabolism but also signaling. The authors observe some fluxes that conflict with the established literature (e.g. glutamate excretion), however the authors do attempt to reconcile this with some experiments and discussion. It may, however, potentially also be warranted to weaken the conclusion somewhat e.g. “most amino acids can be described by transport”. The experimental work seems thorough, however, there is insufficient explanation of the underlying assumptions and data processing as well as improvement potential for the data presentation and simulations that may be addressed by a major revision. See specific comments below.

Response: The authors thank the assessor for a constructive evaluation of the study/manuscript. We recognise that our writing has caused some misunderstanding, particularly regarding glutamate

efflux. We have performed additional extensive analysis and experiments to address all points, remove misunderstanding and provide the requested information. We would like to comment that our statement that metabolism can be disregarded is not absolute, in fact we listed two examples, e.g. glutamate and aspartate, the levels of which is determined by metabolism and transport. Our observations are not in conflict with the literature as we do observe glutamate efflux in the model. The main conclusion of the study is that for most amino acids, transport is significantly faster than metabolism. As a result, their levels are mainly determined by transport and changing the metabolic rate in the model has little impact on their equilibrium levels. For amino acids with very slow transport, particularly aspartate and glutamate, metabolism has to be considered. We have now addressed this in the introduction and conclusion by writing:

“Here we show that this relationship is mainly determined by a combination of transport processes, which generate a stable equilibrium. The main exceptions are glutamate and aspartate, for which transport is slow in many cells, while metabolism is fast.”

We are pleased that the consequences for amino acid signalling are being recognised, which are fields of current enquiry.

Major comments:

The maximum activity (v_{max}) of a transporter depends on both the turnover number (k_{cat}) and the concentration in the membrane ($[E]$). Both the mRNA analysis and the westerns are attempts to quantify differences in $[E]$. This may be sufficient for determining absence/presence of a transporter, however, since in principle k_{cat} could differ by orders of magnitude, it may not be appropriate to compare absolute level of mRNA or protein between different transporters, a value that is considered high for one may be considered low for another. The Human protein atlas (proteinatlas.org) has tabulated mRNA values for a large number of cell lines including A549 and U87-MG, perhaps it would be more informative to compare the level of each transporter to the level in all other cell lines e.g. by its z-score. It is also not made clear exactly how the information from this survey is used in the remaining study, presumably transporters that are considered absent are not considered in the v_{max} study and blocked in the metabolic model, but how are transporters with conflicting evidence treated? Have alternative explanations been considered for instances where there is no detectable mRNA but detectable protein, e.g. issues with antibody specificity? Further, the criteria for the + system in figure 1 is not clearly explained, does it correspond to the 3 different types of measurements? In such case what cutoffs are used? The figure legend also states that high mRNA expression correlates with western blotting, however, no such correlation is calculated or shown. Some transporters are marked as “inducible” with no further explanation, how is this determined?

Response: This is a misunderstanding, we did not use mRNA and western blot analysis to quantify $[E]$. In fact, we state “qualitatively” in the manuscript. These experiments/database searches were performed to catalogue candidates for later analysis. The (+) signs aim to support the qualitative assessment and their meaning is now explained in the figure legend. Most transporters can be described as house-keeping genes but SNAT2 in particular is highly inducible. It is nearly absent when fresh media are added to a culture, but increases its activity when media get depleted.

To clarify we now write: *“To qualitatively survey the amino acid transporter set in each cell line and to identify candidates for transport analysis, we analysed public databases and used RT-PCR to determine the expression of amino acid transporters at the transcriptomic level”.*

We do not want to overemphasize this qualitative assessment by using scores and cut-offs. We have also added further explanations to former Fig. 1, which we have now moved to supplemental data as Fig. S1.

Conflicting evidence was not a problem in our study, it rather alerted us to the possible presence of certain transporters, which we then clarified in the transport analysis. We have carefully evaluated antibodies used in this study and discarded quite a few based on experimental evidence, e.g. incorrect molecular weight or known absence in a particular tissue. The final judgement, what transporters to include in the model rested on the transport experiments. The assessor is correct that there is no strong relationship between k_{cat} and $[E]$ and we have not used any of the mRNA and western blot data to derive such a relationship. Again, the transport experiments provided us with V_{max} data, neither k_{cat} or $[E]$ were required to run the model.

The assumptions underlying the experimental v_{max} determination from figure 2,3,4 are not clear. The concentration used (100uM) is lower than K_m for many of the transporters, so it cannot safely be assumed that all transporters operate at v_{max} in this experiment, it is not clear if this has been considered when deconvoluting the contributions from the different transporters. On the other hand, in table S5 there is a distinction between activity and v_{max} , but it is not stated how v_{max} is calculated from activity. Perhaps v_{max} is here adjusted by the relation between K_m and the substrate concentration in the experiment? However, for some transporters the calculated v_{max} appears to be lower than the activity, could this be due to different units for activity and v_{max} ? There is also color coding in table S5 that is not explained in the legend, additionally it mentions that some values were excluded from the mean, what is this mean value used for and how were the values selected for exclusion?

Response: We appreciate the effort of the assessor to go through the individual datasets. There is a misunderstanding here, which can be readily clarified. In radioactive uptake experiments a concentration $<K_m$ is chosen to maximise the specific activity. The V_{max} is calculated after rearranging the Michaelis-Menten equation.

$$V_{max} = \frac{v (K_m + [S])}{[S]}$$

v is given as nmol amino acid uptake per minute. This explains why the final activity can be below the actual activity, because for some substrate uptakes were measured over 2 or 6 min, and rates were divided by 2 or 6 accordingly.

To address the limited detail on some of the supplementary tables, we have now added a paragraph to the methods section outlining how input parameters were generated and calculated. We have also revised Table S5 (now Table S6) and added explanatory comments.

It seems like an implicit assumption that inhibition is 100% efficient. If this is the case, how are the concentrations of inhibitors determined? What is the reason that some inhibitors are listed in supplementary table S1. E.g. NMDG but not shown in figure 2 and 3, is this because they are assumed to not be present in this particular cell line based on the results in figure 1?

Response: A related comment was made by reviewer 2. For principal reasons inhibition cannot be 100%, realistically $> 95\%$. We have adjusted inhibitors to concentrations that are $20 \times IC_{50}$ where possible. Under these constraints very little transport activity remains to be clarified. We now visualize

this in Fig. 1 and 2 (previously Fig. 2 and 3). Some discrepancies arise because earlier generations of inhibitors, such as BCH and MeAIB are less specific than more recently developed inhibitors.

Where doubts remained, we used RNAi as shown in Fig. S5, but RNAi inhibition is incomplete, as well. We apologise for not explaining NMDG at first usage. NMDG is an ion replacement not an inhibitor. We now write: Ion dependence was analysed by replacing Na^+ with the impermeable organic cation N-Methyl-D-glucamine⁺ (NMDG) or with Li^+ . The catalogue of transporters as shown in Fig.1 is indeed an important step to simplify transporter analysis, but this descriptive information has now been moved to the supplement (Fig. S1).

It is mentioned that nonspecific transport and binding is determined by addition of unlabeled substrate and subtracted from the data. Is this amount negligible compared to the specific transport? Is there any particular benefit of subtracting this value instead of treating it as another factor in the ANOVA?

Response: The background transport is indeed negligible, it would lift all bars by a small amount without adding information. Moreover, the physical nature of background counts is unclear. Most likely it reflects radioactive substrate sitting in gaps between cells that is not easily washed out, or non-specific binding to the surface of the cells. In each case, we want to analyse true uptake of labelled amino acids.

Mathematically, how are the contributions from the different transporters (shown in figure 4) determined? Is this done by summing up the factors in the ANOVA? In such case, perhaps reporting the confidence intervals may be informative for the reader.

Response: The contributions are calculated by subtraction of transport conditions in Fig. 2 and 3 (Now Fig. 1 and 2) to isolate the contributions of a particular transporter. To visualise this process, we have colour-coded Figures 1 and 2 and provide STD in Fig. 3 (previously Fig. 4).

Many cell lines excrete glutamate, alanine and glycine (See [Jain 2012] for a high throughput study of exchange fluxes that includes A549 or [Nilsson 2020] for a quantitative analysis of amino acid excretion). What could be the reason that only uptake is observed in this study? Is this a property specific to the studied cell lines? Are the cells nutrient/glutamine starved prior to the experiment? Or could there potentially be mixing between the medium and cytoplasm which does not constitute a net influx?

Response: This is a misunderstanding. Due to the lack of radioactive amino acids inside the cell (trans 0), we only register the uptake of an amino acid, while the efflux of non-radioactive amino acids remains invisible. The simulation does not have this constraint and shows substantial efflux including glutamate. We now document this in detail in Fig. 5 and 6. We have performed additional experiments and can show that the second glutamate efflux pathway suggested by Nilsson is mediated by ASCT2. This was predicted by simulation (see analytical graph below, which shows negative fluxes for glutamate by ASCT2 (blue) and xCT (red)) and we confirmed it experimentally as shown in Fig. 4 d and e. There is also efflux of many other amino acids through antiporters and controllers as shown in Fig. 5,6 and 7. The misunderstanding may have resulted from former Fig S6a (now S7c), which just shows one particular simulation where uptake dominates due to low initial intracellular amino acid concentrations.

Km values are collected from literature and it is stated that missing Km values are interpolated, this cannot be correct, as interpolation assumes that there is a curve to interpolate. Perhaps the mean or median is used? There seems to be no further mentioning of interpolation in methods or supplementary material. Many of the Km values say '(est)' which I assume means estimated, but it is not clear how these estimations were performed.

Response: Estimation might be the better word than interpolation to determine Km-values where experimental data were missing. For example, if a Km for leucine is known we assumed the Km of isoleucine to be the same. If a trend was known that small amino acids have a higher affinity than large amino acids the Km of larger amino acids was increased proportionally (an approximate interpolation). Methionine often has a similar Km to branched-chain amino acids, not to cysteine. These are indeed estimates, and prone to deviation, but in the absence of better data this limitation is not easily overcome. In the table caption we now write: *"The table shows a curated list of Km-values, which in the opinion of the authors represent the most reliable experimental values. Many studies have not determined the Km of all possible substrates, in which case an estimate (est) was made based on physical relation to known substrate Km-values and general trends of Km-values observed for a particular transporter."*

Using these Km values and v_{max}es the authors construct a metabolic model of amino acid transport based on Michaelis–Menten (MM) based kinetics. Through some optional parameters the model is also equipped to model steady state metabolic fluxes, but since this option does not seem to be used in this study, the presentation of this should perhaps be deferred to the discussion section. The authors opt to solve the model through an iterative approach. It is not made clear why this is preferred over using an ordinary differential equation solver, e.g. `matlabs ode45`.

Response: As the assessor suggests, the modelling can be performed in different ways. We have considered `ode45`, but found this way of modelling more intuitive. For glutamate and glutamine, inclusion of a metabolic conversion was used to improve the fit of the data. We have now added this parameter to Table S4.

Transporters with multiple substrates are handled by re-estimating apparent K_ms at each timestep. It seems like the equation for this (eq2) could have some unintuitive consequences, e.g. if one only considers a single amino acid at a concentration of K_m then it seems like the equation would become $K_{app} = K_m (1 + 1/1) = 2K_m$. Perhaps this is just an issue with the mathematical presentation, but the amino acids should not inhibit themselves.

Response: As the assessor suggests, this misunderstanding arose from the mathematical notation. To clarify we write: “Competition between substrates is incorporated by calculation of an apparent K_M for amino acid (i) competing with other amino acid (a) substrates of the transporter:

$$K_{apps,i} = K_{Mi} \left(1 + \sum \frac{AA_a}{K_{Ma}} \right)$$

To ensure compliance with the 2nd law of thermodynamics the authors set K_m and V_{max} to the same value for uniporters (principle 1). However strictly, it should be sufficient to require that the equilibrium constant (K_{eq}) = 1 for this to hold. From the Haldane relationship ($K_{eq} = (V_f * K_{mP}) / (V_b * K_{mS})$) it follows that $V_b/V_f = K_{mP}/K_{mS}$. So only together with principle 4, that v_{max} is equal for the forward and reverse reaction does it follow that K_{mP} must be equal to K_{mS} . It is also explicitly required (principle 3) that the flux is zero when the metabolite is absent, it is not clear why this does not follow directly from the equations that would involve a multiplication with zero concentration.

Response: As the assessor suggests, the principles underlying the simulation could be phrased in several ways. In our simulation input parameters are K_m , V_{max} and amino acid concentrations, while K_{eq} is an output parameter. We have defined the restrictions for the input parameters and they are validated by correctly computing K_{eq} output values. These K_{eq} values are now listed in Table S5. For this we have run simulations with single transporters to validate that the accumulation conforms to the theoretical value. The Haldane relationship is easily applicable to uniporters, but less so when electrogenic driving forces are involved. The assessor is correct that principle 1 and 4 must be combined to explain $K_{eq}=1$ for a uniporter. Since principle 4 applies to all transporters, we have now listed it as principle 1, followed by the others. We further write: “A couple of restrictions apply to input parameters to avoid violating the 2nd law of thermodynamics”. This has important practical consequences, because inward facing K_m -values are very difficult to measure experimentally. We agree with the assessor that principle 3 is not strictly necessary, rather added as a comment. As a result, we have removed principle 3.

The model predicts amino acid concentrations in good agreement with measured values. However, since the data spans orders of magnitudes, the model could in principle be correct with regards to the typical concentration of each amino acid, but at the same time be incorrect with regards to changes between cell lines, e.g. predicting a 10% increase instead of a 10% decrease would not affect the overall correlation between experimental data and model prediction. A correlation of the change would address this, this type of analysis is performed for myotubules with some success (figure 6E) but should also be performed for A549 vs U87-MG.

Response: This is a valid point that we now address in detail in Fig. 5 and 6 and Fig S8 using experiments similar to those performed with the myotubes. In each case we have plotted the correlation for the absolute concentrations and the changes.

It seems like several of the discussion points, including the proposed functional classification of AA transporters, could be addressed in silico with a sensitivity analysis/metabolic control analysis (MCA) of the model. Such an analysis would provide numerical answers about the contributions of different transporters and parameters (including cell volume) to the model predictions. To facilitate this type of systematic analysis it may be advisable to restructure the source code so that the functional part becomes independent of the UI.

Response: This is an excellent suggestion. We have performed sensitivity analysis by removing individual transporters from the simulation and recording the change to the amino acid concentration. The results support our classification and are now shown in Fig. 7.

Since the model relies on experimentally determined v_{max} values and the metabolite levels predicted for myotubes from transcriptomics data appear to be off by a factor 100 (figure 6E and figure S6F) possibly due to the missing volume estimates, it may be a too strong conclusion that concentrations can be calculated using mRNA levels, however, it appears that the relative levels can be estimated.

Response: We have now performed morphometric measurements of myotube volume, which improved the quality of the data and the fit of the simulation.

Minor comments:

On line 180 it is stated that amino acid transport is “governed by” MM, but MM is just a simplified model, perhaps it would be more correct to write “can be described by”.

Response: As suggested we write: “Amino acid transport can be described by Michaelis-Menten kinetics after incorporating electrochemical driving forces”

In figure 1 it is not clear why the transporter names are repeated for each cell line.

Response: Due to the high line density, we feel that this is easier for the reader. The figure has been moved to the supplement, because of additional new figures.

When error bars are shown does this represent SEM or STD? It is becoming increasingly common to show the data points along with the barplot when the N is sufficiently small.

Response: Apologies for this oversight, we are stating STD now in figure legends.

The legends of figure 2 and 3 state that the figures show a deconvolution, however the figure only shows the data, it is figure 4 that shows the deconvolution.

Point taken, we are now using “Discrimination” in figures 2 and 3 (now figure 1 and 2) and “Deconvolution” in figure 4 (now figure 3).

The number of letters above the bars in figure 2 and 3 do not always match up with the bars e.g. in 2F.

Thanks for spotting, this has been amended as suggested.

A 3d pichart, as in figure 4 is not a usual way to convey scientific data.

A change to stacked bar graphs was also recommended by reviewer 2 and has been made.

In figure 4 all transporters are named by the convention “[Amino acid] Transport” apart from alanine that is named “Alanine transport”.

Thanks for spotting, we have amended this as suggested.

Technically it is not sufficient to show a single instance of an equilibrium point (figure S6A) to show that a system is “inherently stable”. However, from the formulation it does seem very probable that it is.

Response: This is a valid point and discussed in the manuscript. Former Figure S6A (now S7C) stands as an example and indeed most simulations come to a stable equilibrium. We now write: “This was indeed the case and an example is shown in Fig. S7c”. As discussed, EAAT-type glutamate transporter never reach equilibrium, but this is counteracted by low activity, metabolism and efflux.

In the introduction it is assumed that uniporters equilibrate cytosolic and plasma amino acid concentrations, however, in principle a uniporter could provide net transport if it consumes ATP in the process, e.g. ABC transporters.

Response: We have used the most common classification of transporter mechanisms, which is based first on energy source: Primary active transporters are driven by ATP, secondary active transporters are driven by substrate and ion gradients. Secondary active transporters are then subdivided into uniporters, symporters and antiporters. ABC transporters are largely unidirectional, because of the free energy associated with the hydrolysis of ATP, but generally are not referred to as uniporters. To avoid confusion, we now write: “Amino acid transport in mammalian cells is mediated by more than 60 different secondary active transporters such as uniporters, symporters and antiporters.”

Several abbreviations are not defined at first use, e.g. for JDFC.

Response: JDFC is a lab acronym referring to the student who wrote the script (Jade Vennitti, 2nd author) and some of her aspirations. It is a way of paying tribute to her efforts, like naming plasmids or genes. We have added definitions for mTORC1 and GCN2/ATF4.

Table S4 is not a table.

We thought it would be easier to find if given as a “table”, as suggested we have renamed this section into List S4.

It is stated in the manual that only matlab 2019, not 2020 is not supported. It is possible to get most of the code to run under matlab 2020 by adding the line `[variable] = arrayfun(@(x) str2double(x), [variable]);`

We appreciate this suggestion and work on the compatibility of the script.

In figures similar to Figure S6B, it is useful to also show the line of identity.

To show two lines would be confusing to some readers, but we have included the quadrants in the analytical graphs, which helps to estimate the goodness of fit. All axes have the same scale in both dimensions.

In figure 6E, why is the log scale in base 2 instead of 10?

Response: The log 2 scale provides more convenient subdivisions of the scale. The graph looks the same using log 10.

Direct quotes of other work, as done in the introduction, is not common in natural science, paraphrasing and citing is often sufficient.

Response: We thank the reviewer for the suggestion and have removed the quote. The main point here was to highlight the significance of the problem.

REVIEWERS' COMMENTS

Reviewer #1 (Remarks to the Author):

I am totally satisfied by the actions taken by the authors in the new version of the manuscript. Similarly, I fully agree, with the explanations and changes made in the manuscript in relation to the points raised by the other reviewers. I refer myself to questions related to amino acid transport (estimations of K_m values for substrates in transporters not reported in the literature, estimation of the functional activity of transporters based on estimations of V_{max} , main use of inhibitors instead of genetic ablation of transporters for most of the cases, etc ...).

After this cycle of revision, the paper has improved, including the change of title as suggested by one of the reviewers.

I think this is a great and important paper to understand the role of the different set of transporters operating in the studied cells.

I fully recommend publishing this paper in its present state.

Reviewer #2 (Remarks to the Author):

This is a generally excellent manuscript which is a great fit for the journal. That said, there are some details that should be cleaned up before publication:

1. Abstract: the order uniporters, symporters, and antiporters, should align with the order listing the functional activities.
2. Introduction: the introduction does not provide an easy overview of amino acid transport, which does not come until the last figure. An introductory figure covering the main transporters of interest (with both common and SLC names) would make the paper much more readable.
3. Since the authors do not evaluate all amino acids thoroughly for rates of transport versus use, instead of claiming that aspartate and glutamate are the main exceptions (more use than transport), they should just point them out as exceptions, as I believe that there are other exceptions at least in vivo.
4. Line 207: Where does the 0.5 come from in the exponent?
5. Line 439: Amino acid metabolism does not cease during fasting. There is no analysis of fasting (or other in vivo biology) in this paper and existing literature shows active amino acid metabolism during fasting.
6. Line 443: I do not believe the explanation of the glycine cancer cell results is correct--this has to do with what happens before and after serine is consumed from the media, not transport.
7. Line 462: The manuscript is restricted to cell culture studies, therefore it does not establish a relationship between plasma and intracellular amino acid levels.

Reviewer #3 (Remarks to the Author):

With this revision, most comments have been satisfactorily addressed, remaining comments may be addressed in a minor revision.

Major comments

Regarding $[E] \times [K_{cat}] = V_{max}$, it is clear that v_{max} was determined experimentally for the transporters that were included in the study, however, a qualitative assessment of expression was used to decide which transporters to include. This assessment implicitly assumes that if $[E]$ is sufficiently small then $V_{max} \sim 0$ and the transporter can be excluded from further analysis (an appropriate assumption). However, since $[K_{cat}]$ may differ widely between transporters, the cut off for

when [E] is sufficiently small will also differ. This means that absolute mRNA levels (as here are used as a proxy for [E]) will not be comparable between different transporters; an mRNA level that is low enough to be considered absent for one gene, may be the typical expression level for another, e.g. SLC7A7S is expressed at around 300 NX, SLC7A14 at around 30 NX and SLC7A10 around 3 NX (proteinatlas.org). Rather than comparing absolute values of mRNA it would be more relevant compare each gene against a typical expression for that gene, as can be found in repositories, this may perhaps resolve some of the apparent conflicting results between mRNA and western.

It would be appropriate to more clearly document how each estimated K_m value is estimated. The model and/or parameters may be reused by others and they may not be aware of the different interpolation techniques of k_m vs size (how was btw size defined? molecular weight?) and similarities between different subclasses of amino acids. A more transparent description of the parameter estimation may make it more clear that the parameter estimation was not influenced by the data fits.

Thermodynamics require $K_{eq}=1$ for uniporters, however, this does not imply that $V_f=V_b$ and $K_{mP}=K_{mS}$, any combination of parameters that satisfy $(V_f * K_{mP})/(V_b * K_{mS})=1$ would be permissible, see E. Bosdriesz 2018 et. al. (Second paragraph in results and Supplementary Information page 3) for a more thorough discussion. Restrictions 1 and 2 are sensible, but not necessary, assumptions. To clarify this it, "restrictions" may perhaps be replaced by "assumptions".

Minor comments

In the revised version of the manuscript it is stated that "metabolism ceases" in the fasting state. However, metabolism does not cease as long as an organism is alive.

It may be worth considering if some of the material in discussion that relates to sensitivity analysis perhaps may now be considered results.

References

E. Bosdriesz, et al., Low affinity uniporter carrier proteins can increase net substrate uptake rate by reducing efflux. *Sci. Rep.* 8, 5576 (2018).

Reviewer #1

I am totally satisfied by the actions taken by the authors in the new version of the manuscript. Similarly, I fully agree, with the explanations and changes made in the manuscript in relation to the points raised by the other reviewers. I refer myself to questions related to amino acid transport (estimations of K_m values for substrates in transporters not reported in the literature, estimation of the functional activity of transporters based on estimations of V_{max} , main use of inhibitors instead of genetic ablation of transporters for most of the cases, etc ...).

After this cycle of revision, the paper has improved, including the change of title as suggested by one of the reviewers.

I think this is a great and important paper to understand the role of the different set of transporters operating in the studied cells.

I fully recommend publishing this paper in its present state.

Response: No response required, we thank the reviewer for his positive and encouraging evaluation.

Reviewer #2

This is a generally excellent manuscript which is a great fit for the journal. That said, there are some details that should be cleaned up before publication:

1. Abstract: the order uniporters, symporters, and antiporters, should align with the order listing the functional activities.

Response: Uniporters, symporters and antiporters do not fully align with controllers, loaders and harmonizers. For instance, CAT1 is a uniporter and a loader and SNAT5 is a symporter and a controller. We would have used 'respectively' if the alignment was complete. To avoid misunderstanding we now write "This work explains amino acid homeostasis at the systems-level, through a combination of secondary active transporters, functionally acting as loaders, harmonizers and controller transporters to generate a stable equilibrium of all amino acid concentrations."

2. Introduction: the introduction does not provide an easy overview of amino acid transport, which does not come until the last figure. An introductory figure covering the main transporters of interest (with both common and SLC names) would make the paper much more readable.

Response: This is a good suggestion, the scheme is now Fig. 1

3. Since the authors do not evaluate all amino acids thoroughly for rates of transport versus use, instead of claiming that aspartate and glutamate are the main exceptions (more use than transport), they should just point them out as exceptions, as I believe that there are other exceptions at least in vivo.

This could very well be the case if transport activity for a particular amino acid would be very low, while metabolism fast. Glutamate in liver is the prime example, but that would be covered by our statement. To avoid overarching statements we write "The exceptions in many cell lines are glutamate and aspartate, for which transport is typically slow, while metabolism is fast."

4. Line 207: Where does the 0.5 come from in the exponent?

Well spotted, it was a mistake in the original submission, which we corrected in the revised version. In the thermodynamic equation $[exp(-zF\Delta\Psi/RT)]$ the electric component provides driving force for a ~10 fold

*accumulation of the substrate. In the rate equations we multiply the forward transport by β (i.e. *10) and the reverse transport by $1/\beta$ (i.e. *0.1). This gives a 100-fold difference between forward and reverse fluxes and would result in a ~100-fold accumulation. Using half of the electric component in each case multiplies the forward rate by 3.33 and the reverse flux by 0.3 resulting in a 10-fold difference. The equivalence was demonstrated in Table S5. Mechanistically this can be rationalised by the structural observation that binding sites are in the centre of the membrane thus only part of the electrical field affects the translocation of the substrate in each direction.*

5. Line 439: Amino acid metabolism does not cease during fasting. There is no analysis of fasting (or other in vivo biology) in this paper and existing literature shows active amino acid metabolism during fasting.

Completely agreed, in fact I tell the same to students in my biochemistry class, only to find out that I made the same mistake in my own writing. We now use “metabolism slows down, particularly that of essential amino acids”. This excludes non-essential amino acids, which contribute to gluconeogenesis during fasting.

6. Line 443: I do not believe the explanation of the glycine cancer cell results is correct--this has to do with what happens before and after serine is consumed from the media, not transport.

Response: This is probably a misunderstanding of our argument. The point we wanted to make is that the relationship between intracellular and extracellular amino acids generated by transporters automatically results in efflux when there is an excess of amino acids and automatically results in uptake when there is net demand. To avoid potential misunderstanding we write: “The constant relationship between extracellular and intracellular amino acid concentrations also explains why glycine is released in slowly growing cancer cell lines, because there is excess glycine while it is consumed in fast growing cell lines, when net demand produces an imbalance in the cytosol. We included this section and reference to address a comment made by reviewer 3.

7. Line 462: The manuscript is restricted to cell culture studies, therefore it does not establish a relationship between plasma and intracellular amino acid levels.

Response: We have rephrased to “a relationship between extracellular and intracellular amino acid levels”. We do very much think that the same principles apply in vivo and that a knowledge of transporter expression in a particular cell type (for example using single-cell transcriptomics) in vivo would allow a prediction of cytosolic amino acid levels based on plasma amino levels with restrictions applying to amino acids with low permeability as discussed above.

Reviewer #3

With this revision, most comments have been satisfactorily addressed, remaining comments may be addressed in a minor revision.

Regarding $[E] \times [K_{cat}] = V_{max}$, it is clear that v_{max} was determined experimentally for the transporters that were included in the study, however, a qualitative assessment of expression was used to decide which transporters to include. This assessment implicitly assumes that if $[E]$ is sufficiently small then $V_{max} \sim 0$ and the transporter can be excluded from further analysis (an appropriate assumption). However, since $[K_{cat}]$ may differ widely between transporters, the cut off for when $[E]$ is sufficiently small will also differ. This means that absolute mRNA levels (as here are used as a proxy for $[E]$) will not be comparable between different transporters; an mRNA level that is low enough to be considered absent for one gene, may be the typical expression level for another, e.g. SLC7A7S is expressed at around 300 NX, SLC7A14 at around 30 NX and SLC7A10 around 3 NX (proteatlas.org). Rather than comparing absolute values of mRNA it would be more relevant compare each gene against a typical expression for that gene, as can be found in repositories, this may perhaps resolve some of the apparent conflicting results between mRNA and western.

Response: There is a possibility that we may have overlooked a transport component of a transporter whose mRNA is well translated, has a high K_{cat} and is difficult to detect by transport measurements, such as a low-affinity controller transporter. As a potential candidate we have checked normalised RNA levels for SLC38A5 in A549 and U87-MG and found the same results as obtained by our analysis. To address this comment we write: “Moreover, transporters with high catalytic rates and with low transcript copy numbers which are efficiently translated, may be underestimated or overlooked.”

It would be appropriate to more clearly document how each estimated K_m value is estimated. The model and/or parameters may be reused by others and they may not be aware of the different interpolation techniques of k_m vs size (how was btw size defined? molecular weight?) and similarities between different subclasses of amino acids. A more transparent description of the parameter estimation may make it more clear that the parameter estimation was not influenced by the data fits.

Response: In Table S3 we now indicate when equality of K_m 's was assumed between different amino acids or where experimental data were used for estimates. In many studies K_m values were measured for key substrates, but competitive inhibition was measured for a wider variety at a single concentration. The differences in the ability to compete with a key substrate was used where possible to estimate the K_m and this is now listed in the supplemental table.

Thermodynamics require $K_{eq}=1$ for uniporters, however, this does not imply that $V_f=V_b$ and $K_{mP}=K_{mS}$, any combination of parameters that satisfy $(V_f * K_{mP})/(V_b * K_{mS})=1$ would be permissible, see E. Bosdriesz 2018 et. al. (Second paragraph in results and Supplementary Information page 3) for a more thorough discussion. Restrictions 1 and 2 are sensible, but not necessary, assumptions. To clarify this it, “restrictions” may perhaps be replaced by “assumptions”.

Response: We generally agree, but this would require modelling a 4-state carrier system, for which intracellular K_m -values have to be known (as in the cited reference). This is not the case for all but a few amino acid transporters. The advantage of the state-state equations used here is that we only need to know extracellular K_m -values and V_{max} to compute transport activity, because the intracellular K_m -values have to be the same, with the exception of antiporters. We are aware that in situ Haldane relationships exist where K_{mP} and K_{mS} are different. Our simulation is a practical simplification because intracellular K_m values are largely unknown for amino acid transporters. To clarify we write “A couple of conditions apply to input parameters to avoid violating the 2nd law of thermodynamics in the simulations.”

Minor comments

In the revised version of the manuscript it is stated that “metabolism ceases” in the fasting state. However, metabolism does not cease as long as an organism is alive.

Response: Completely agreed, in fact I tell the same to students in my biochemistry class, only to find out that I made the same mistake in my own writing. Corrected as requested.

It may be worth considering if some of the material in discussion that relates to sensitivity analysis perhaps may now be considered results.

Response: We would prefer to leave this in the discussion, because we are introducing the terms loader, harmonizer and controller in this section and the sensitivity analysis supports these classifications.

References

E. Bosdriesz, et al., Low affinity uniporter carrier proteins can increase net substrate uptake rate by reducing efflux. *Sci. Rep.* 8, 5576 (2018).